# VIDEODIRECTORGPT: Consistent Multi-Scene Video Generation via LLM-Guided Planning

**Han Lin    Abhay Zala    Jaemin Cho    Mohit Bansal**
UNC Chapel Hill
{hanlincs, aszala, jmincho, mbansal}@cs.unc.edu
https://videodirectorgpt.github.io/

## Abstract

Recent text-to-video (T2V) generation methods have seen significant advancements. However, the majority of these works focus on producing short video clips of a single event (i.e., single-scene videos). Meanwhile, recent large language models (LLMs) have demonstrated their capability in generating layouts and programs to control downstream visual modules. This prompts an important question: *can we leverage the knowledge embedded in these LLMs for temporally consistent long video generation?* In this paper, we propose VIDEODIRECTORGPT, a novel framework for consistent multi-scene video generation that uses the knowledge of LLMs for video content planning and grounded video generation. Specifically, given a single text prompt, we first ask our video planner LLM (GPT-4) to expand it into a '*video plan*', which includes the scene descriptions, the entities with their respective layouts, the background for each scene, and consistency groupings of the entities. Next, guided by this *video plan*, our video generator, named Layout2Vid, has explicit control over spatial layouts and can maintain temporal consistency of entities across multiple scenes, while being trained only with image-level annotations. Our experiments demonstrate that our proposed VIDEODIRECTORGPT framework substantially improves layout and movement control in both single- and multi-scene video generation and can generate multi-scene videos with consistency, while achieving competitive performance with SOTAs in open-domain single-scene T2V generation. Detailed ablation studies, including dynamic adjustment of layout control strength with an LLM and video generation with user-provided images, confirm the effectiveness of each component of our framework and its future potential.

## 1  Introduction

Text-to-video (T2V) generation has achieved rapid progress following the success of text-to-image (T2I) generation. Most works in T2V generation focus on producing short videos (e.g., 16 frames at 2fps) from the given text prompts (Wang et al., 2023b; He et al., 2022; Ho et al., 2022; Singer et al., 2023; Zhou et al., 2022). Recent studies on long video generation (Blattmann et al., 2023b; Yin et al., 2023; Villegas et al., 2023; He et al., 2023) aim at generating long videos of a few minutes with holistic visual consistency. Although these works could generate longer videos, the generated videos often display the continuation or repetitive patterns of a single action (e.g., driving a car) instead of transitions and dynamics of multiple changing actions/events (e.g., five steps about how to make caraway cakes). Meanwhile, large language models (LLMs) (Brown et al., 2020; OpenAI, 2023; Touvron et al., 2023a;b; Chowdhery et al., 2022) have demonstrated their capability in generating layouts and programs to control visual modules (Dídac et al., 2023; Gupta & Kembhavi, 2023), especially image generation models (Cho et al., 2023b; Feng et al., 2023). This raises an interesting question: *Can we leverage the knowledge embedded in these LLMs for planning consistent multi-scene video generation?*

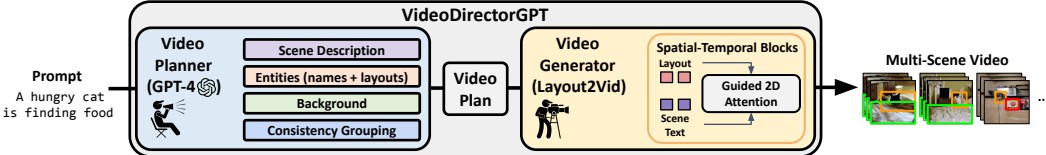

Figure 1: **VIDEODIRECTORGPT framework.** First, we employ **GPT-4 as a video planner** to craft a multi-component *video plan*. Next, we utilize **Layout2Vid**, a grounded video generation module, to render multi-scene videos with layout and consistency control.

In this work, we introduce **VIDEODIRECTORGPT**, a novel framework for consistent multi-scene video generation. As illustrated in Fig. 1, VIDEODIRECTORGPT decomposes the T2V generation task into two stages: **video planning** and **video generation**. For the first video planning stage (see Fig. 1 blue part), we employ an LLM to generate a *video plan*, which is an overall plot of the video with multiple scenes, each consisting of a text description of the scene and entity names/layouts, and a background. It also consists of consistency groupings of specific entities that re-appear across scenes. For the second video generation stage (see Fig. 1 yellow part), we introduce Layout2Vid, a novel grounded video generation module that generates multi-scene videos from the *video plan*. Our framework provides the following strengths: (1) employing an LLM to write a *video plan* that guides the generation of videos with multiple scenes from a single text prompt, (2) layout control in video generation by only using image-level layout annotations, and (3) generation of visually consistent entities across multiple scenes.

In the first stage, video planning (Sec. 3.1), we employ an LLM (e.g., GPT-4 (OpenAI, 2023)) as a video planner to generate a *video plan*, a multi-component video script with multiple scenes to guide the downstream video synthesis process. Our *video plan* consists of four components: (1) multi-scene descriptions, (2) entities (names and their 2D bounding boxes), (3) background, and (4) consistency groupings (scene indices for each entity indicating where they should remain visually consistent). We generate the *video plan* in two steps by prompting an LLM with different in-context examples. In the first step, we expand a single text prompt into multi-step scene descriptions with an LLM, where each scene is described with a text description, a list of entities, and a background (see Fig. 2 blue part for details). We also prompt the LLM to generate additional information for each entity (e.g., color, attire, etc.), and group together entities across frames and scenes, which will help guide consistency during the video generation stage. In the second step, we expand the detailed layouts of each scene with an LLM by generating the bounding boxes of the entities per frame, given the list of entities and scene description. This overall '*video plan*' guides the downstream video generation.

In the second stage, video generation (Sec. 3.2), we introduce Layout2Vid, a grounded video generation module to render videos based on the generated *video plan* (see yellow part of Fig. 2). For the grounded video generation module, we build upon ModelScopeT2V (Wang et al., 2023b), an off-the-shelf T2V generation model, by freezing its original parameters and adding spatial/consistency control of entities through a small set of trainable parameters (13% of total parameters) through the gated-attention module (Li et al., 2023). This enables our Layout2Vid to be trained solely on layout-annotated images, thus bypassing the need for expensive training on annotated video datasets. To preserve the identity of entities across scenes, we use shared representations for the entities within the same consistency group. We also propose to use a joint image+text embedding as entity grounding conditions which we find more effective than the existing text-only approaches (Li et al., 2023) in entity identity preservation (see Appendix H). Overall, our Layout2Vid avoids expensive video-level training, improves the object layout and movement control, and preserves objects cross-scene temporal consistency.

We conduct experiments on both single-scene and multi-scene video generation. Experiments show that our VIDEODIRECTORGPT demonstrates better layout skills (object, count, spatial, scale) and object movement control compared with ModelScopeT2V (Wang et al., 2023b) as well as more recent video generation models including AnimateDiff (Guo et al., 2023), I2VGen-XL (Zhang et al., 2023), and SVD (Blattmann et al., 2023a). In addition,

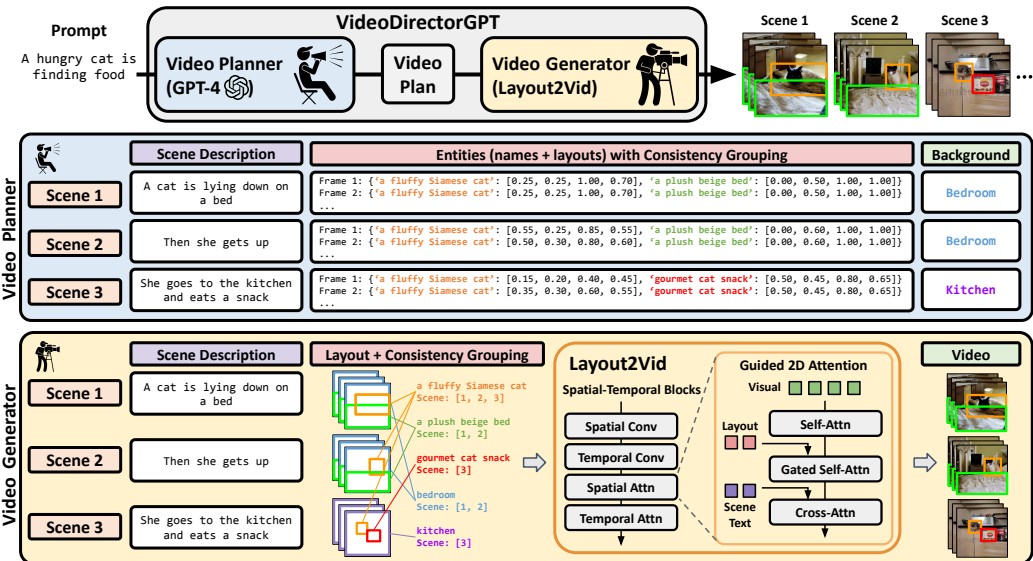

Figure 2: **VIDEODIRECTORGPT details.** **1st stage**: we employ the **LLM as a video planner** to craft a *video plan*, which provides an overarching plot for multi-scene videos. **2nd stage**: we utilize **Layout2Vid**, a grounded video generation module, to render videos based on the *video plan*. (Sec. 3.2).

VIDEODIRECTORGPT is capable of generating multi-scene videos with visual consistency across scenes, and competitive with SOTAs on single-scene open-domain T2V generation (Sec. 5.1 and Sec. 5.2). Detailed ablation studies, including dynamic adjustment of layout control strength, video plans generated from different LLMs, and video generation with user-provided images confirm the effectiveness and capacity of our framework (Sec. 5.3 and Appendix H). Our main contributions can be summarized as follows:

- A new T2V generation framework VIDEODIRECTORGPT with two stages: video content planning and grounded video generation, which is capable of generating a multi-scene video from a single text prompt.
- We employ LLMs to generate a multi-component '*video plan*' with detailed scene descriptions, entity layouts, and entity consistency groupings to guide video generation.
- We introduce Layout2Vid, a novel grounded video generation module, which brings together image/text-based layout control ability and entity-level temporal consistency. Our Layout2Vid can be trained efficiently using only image-level layout annotations.
- Empirical results demonstrate that our framework can accurately control object layouts and movements, and generate temporally consistent multi-scene videos.

## 2 Related Works

**Text-to-video generation.** Training a T2V generation model from scratch is computationally expensive. Recent work often leverages pre-trained T2I generation models such as Stable Diffusion (Rombach et al., 2022) by fine-tuning them on text-video pairs (Wang et al., 2023b; Blattmann et al., 2023b). While this warm-start strategy enables high-resolution video generation, it comes with the limitation of only generating short video clips, as T2I models lack the ability to maintain consistency through long videos. On the other hand, recent works on long video generation (Blattmann et al., 2023b; Yin et al., 2023; Villegas et al., 2023; He et al., 2023) aim at generating long videos of several minutes. However, the generated videos often display the continuation or repetitive actions instead of transitions of multiple actions/events. In contrast, our Layout2Vid infuses layout control and multi-scene temporal consistency into a pretrained T2V generation model via data and parameter-efficient training while preserving its original visual quality.

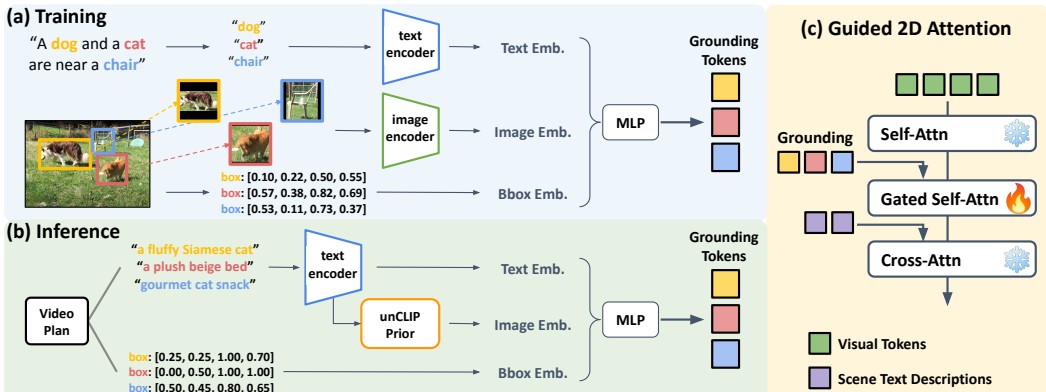

Figure 3: **Illustration of Layout2Vid. (a) Training.** Our Layout2Vid is efficiently trained on image-level bounding box annotations. We construct entity grounding tokens with an MLP that takes text, image, and bounding box embeddings as inputs. **(b) Inference.** we obtain image embeddings of entities from their text descriptions with unCLIP Prior. To ensure temporal consistency, we use the same entity embeddings across scenes. **(c) Guided 2D Attention.** We modulate the visual representation with grounding tokens and text tokens.

**Bridging text-to-image generation with layouts.** To achieve interpretable and controllable generation, a line of research decomposes the T2I generation task into two stages: text-to-layout generation, and layout-to-image generation. While early models train the layout generation module from scratch (Hong et al., 2018; Tan et al., 2019; Li et al., 2019a; Liang et al., 2022), recent methods employ pretrained LLMs to leverage their knowledge in generating image layouts from text (Cho et al., 2023b; Feng et al., 2023; Qu et al., 2023). To the best of our knowledge, our work is the first to utilize LLMs to generate structured video plans from text, enabling accurate and controllable long video generation.

## 3  VIDEODIRECTORGPT

### 3.1  Video Planning: Video Plans with LLMs

**Video plan.** As illustrated in the blue part of Fig. 2, GPT-4 (OpenAI, 2023) acts as a planner, providing a detailed *video plan* to guide the video generation. Our *video plan* has four components: (1) **multi-scene descriptions**: a sentence describing each scene, (2) **entities**: names and their 2D bounding boxes, (3) **background**: text description of the location of each scene, and (4) **consistency groupings**: scene indices for each entity indicating where they should remain visually consistent. The *video plan* is generated in two steps by prompting GPT-4 independently. The input prompt template for each step is displayed in Appendix C.

**Step 1: Generating multi-scene descriptions, entity names, and consistency groupings.** In the first step, we employ GPT-4 to expand a single text prompt into a multi-scene *video plan*. Next, we group entities and backgrounds that appear across different scenes using an exact match. For instance, if the 'chef' appears in scenes 1-4 and 'oven' only appears in scene 1, we form the entity consistency groupings as {chef:[1,2,3,4], oven:[1]}. In the subsequent video generation stage, we use the shared representations for the same entity consistency groups to maintain their temporally consistent appearances (see Sec. 3.2).

**Step 2: Generating entity layouts for each scene.** In the second step, we generate a list of bounding boxes for the entities in each frame based on the entities and the scene description. For each scene, we produce layouts for 9 frames, then linearly interpolate to gather information for denser frames (*e.g.*, 16 frames per scene). In line with VPGen (Cho et al., 2023b), we adopt the $[x_0, y_0, x_1, y_1]$ format for bounding boxes, where each coordinate is normalized to fall within the range [0,1]. For in-context examples, we present 0.05 as the minimum unit for the bounding box, equivalent to a 20-bin quantization.

### 3.2 Video Generation: Generating Videos from Video Plans with Layout2Vid

**Preliminaries.** Given a $T$ frame video $x \in \mathbb{R}^{T \times 3 \times H \times W}$ with video caption $c$, we first uses an autoencoder to encode the video into latents $z = \mathcal{E}(x) \in \mathbb{R}^{T \times C \times H' \times W'}$, where $C = 4$ represents the number of latent channels, and $H' = H/8$, $W' = W/8$ represent the spatial dimension of the latents. In Latent diffusion model (LDM) (Rombach et al., 2022), the forward process is a fixed diffusion process which gradually adds noise to the latent variable $z$: $q(z_t|z_{t-1}) = N(z_t; \sqrt{1 - \beta_t}z_{t-1}, \beta_t I)$, where $\beta_t \in (0,1)$ is the variance schedule with $t \in \{1, ..., T\}$. The reverse process gradually produces less noisy samples $z_{T-1}, z_{T-2}, ..., z_0$ starting from $z_T$ through a learnable denoiser model $\epsilon_\theta$. With t as a timestep uniformly sampled from $\{1, ..., T\}$, the training objective of our Layout2Vid can be expressed as: $\min_\theta L_{\text{LDM}} = E_{z, \epsilon \sim N(0, I), t} \| \epsilon - \epsilon_\theta(z_t, t, c) \|_2^2$.

**Layout2Vid: Layout-guided T2V generation.** We implement Layout2Vid by integrating layout control capability into ModelScopeT2V (Wang et al., 2023b), a public T2V generation model based on Stable Diffusion (Rombach et al., 2022). The diffusion UNet in ModelScopeT2V consists of a series of spatio-temporal blocks, each containing four modules: spatial convolution, temporal convolution, spatial attention, and temporal attention. Compared with ModelScopeT2V, our Layout2Vid enables layout-guided video generation with explicit spatial control over a list of entities represented by their bounding boxes, as well as visual and text content. We build upon the 2D attention to create the guided 2D attention (see Fig. 9 in Appendix D). As shown in Fig. 3 (c), the guided 2D attention takes two conditional inputs to modulate the visual latent representation: (1) grounding tokens, conditioned with gated self-attention (Li et al., 2023), and (2) text tokens that describe the current scene, conditioned with cross-attention.

**Temporally consistent entity grounding with image+text embeddings.** While previous layout-guided T2I generation models commonly used only the CLIP text embedding for layout control (Li et al., 2023; Yang et al., 2023), we use the CLIP image embedding in addition to the CLIP text embedding for entity grounding. We demonstrate in our ablation studies (see Appendix H) that this method is more effective than using either text-only or image-only grounding. As depicted below, the grounding token for the $i^{th}$ entity, $h_i$, is a 2-layer MLP which fuses CLIP image embeddings $f_{\text{img}}(e_i)$, CLIP text embeddings $f_{\text{text}}(e_i)$, and Fourier features (Mildenhall et al., 2021) of the bounding box $l_i = [x_0, y_0, x_1, y_1]$. We use learnable linear projection layers, denoted as $P_{\text{img/text}}$, on the visual/text features, which we found helpful for faster convergence during training.

$$h_i = \text{MLP}(P_{\text{img}}(f_{\text{img}}(e_i)), P_{\text{text}}(f_{\text{text}}(e_i)), \text{Fourier}(l_i))$$

The training and inference procedure of our Layout2Vid are presented in Fig. 3 parts (a) and (b) respectively. During training, the image embeddings $f_{\text{img}}(e)$ are obtained by encoding the image crop of the entities with CLIP image encoder. During inference, since all the inputs are in text format (e.g., from the *video plan*), we employ Karlo (Lee et al., 2022), a public implementation of unCLIP Prior (Ramesh et al., 2022) to transform the CLIP text embedding into its corresponding CLIP image embedding. Moreover, our image embeddings can also be obtained from user-provided exemplars during inference by simply encode the images with the CLIP image encoder.

**Parameter and data-efficient training.** During training, we only update the parameters of the guided 2D attention (13% of total parameters) to inject layout guidance capabilities into the pretrained T2V backbone while preserving its original video generation capabilities. Such strategy allows us to efficiently train the model with only image-level layout annotations, while still equipped with multi-scene temporal consistency via shared entity grounding tokens. Training and inference details are shown in Appendix D.

## 4 Experimental Setup

**Evaluated models.** We compare our VIDEODIRECTORGPT with a total of 9 text/image-to-video generation models (see Sec. 5.1 and Appendix F for baseline model details).

**Prompts for single-scene video generation.** For single-scene videos, we (1) evaluate layout control via VPEval Skill-based prompts (Cho et al., 2023b), (2) assess object dynamics through ActionBench-Direction prompts adapted from ActionBench-SSV2 (Wang et al., 2023c), and (3) examine open-domain video generation using MSR-VTT and UCF-101 (Xu et al., 2016; Soomro et al., 2012). We introduce ActionBench-Direction prompts by sampling video captions from ActionBench-SSV2 (Wang et al., 2023c) and balancing the distribution of movement directions. Prompt preparation details are presented in Appendix F.

**Prompts for multi-scene video generation.** For multi-scene video generation, we experiment with (1) a list of sentences describing events – ActivityNet Captions (Krishna et al., 2017) and Coref-SV prompts based on Pororo-SV (Li et al., 2019b), and (2) a single sentence from which models generate multi-scene videos – HiREST (Zala et al., 2023). Coref-SV is a new multi-scene text description dataset that we propose to evaluate the visual consistency of objects across multi-scene videos. Prompt preparation details are given in Appendix F.

**Automated evaluation metrics.** Following previous works (Hong et al., 2022; Wu et al., 2022b; Wang et al., 2023b), we use FID (Heusel et al., 2017), FVD (Unterthiner et al., 2019), and IS (Salimans et al., 2016) scores as video quality metrics, and CLIPSIM (Wu et al., 2021) score for text-video alignment. Given that CLIP fails to faithfully capture detailed semantics such as spatial relations, object counts, and actions in videos (Otani et al., 2023; Cho et al., 2023a;b; Hu et al., 2023), we further propose the use of the following fine-grained metrics:

- **VPEval accuracy**: we employ it for the evaluation of VPEval Skill-based prompts (object, count, spatial, scale), which is based on running skill-specific evaluation programs that execute relevant visual modules (Cho et al., 2023b). Since the VPEval accuracy described above does not cover temporal information, we propose a metric that takes into account temporal information as well as spatial layouts for ActionBench-Direction prompts.

- **Object movement direction accuracy**: we introduce this new metric to evaluate ActionBench-Direction prompts, which takes both temporal information and spatial layouts into consideration. Firstly, we assess whether the target objects move in the direction described in the prompts. The start/end locations of objects are obtained by detecting objects with GroundingDINO (Liu et al., 2023) on the first/last video frames. We then evaluate whether the $x$ (for movements left or right) or $y$ (for movements up or down) coordinates of the objects have changed correctly and assign a binary score of 0 or 1. For instance, given the prompt "pushing a glass from left to right" and a generated video, we identify a 'glass' in both the first and last video frames. We assign a score of 1 if the $x$-coordinate of the object increases by the last frame; otherwise, assign a score of 0.

- **Multi-scene object temporal consistency**: we propose this new metric for the evaluation of ActivityNet Captions and Coref-SV, which measures the consistency of the visual appearance of a target object across different scenes. We also introduce a new metric to measure the consistency of the visual appearance of a target object across different scenes. Specifically, we first detect the target object using GroundingDINO from the center frame of each scene video. Then, we extract the CLIP (ViT-B/32) image embedding from the crop of the detected bounding box. We calculate the multi-scene object consistency metric by averaging the CLIP image embedding similarities across all adjacent scene pairs: $\frac{1}{N} \sum_{n=1}^{N-1} cos(\text{CLIP}_n^{\text{img}}, \text{CLIP}_{n+1}^{\text{img}})$, where $N$ is the number of scenes, $cos(\cdot, \cdot)$ is cosine similarity, and $\text{CLIP}_n^{\text{img}}$ is the CLIP image embedding of the target object in $n$-th scene.

**Human evaluation.** We conduct a human evaluation on the multi-scene videos generated by both our VIDEODIRECTORGPT and ModelScopeT2V on the Coref-SV dataset. Since we know the target entity and its co-reference pronouns in the Coref-SV prompts, we can compare the temporal consistency of the target entities across scenes. We evaluate the human preference between videos from two models in each category of Quality, Text-Video Alignment, and Object Consistency. See Appendix F for setup details.

**Step-by-step error analysis.** We conduct an error analysis at each step of our single-sentence to multi-scene video generation pipeline on the HiREST dataset. We analyze the generated multi-scene text descriptions, layouts, and entity consistency groupings to evaluate our video planning stage, and examine the final video to evaluate the video generation stage.

# 5 Results and Analysis

## 5.1 Single-Scene Video Generation

**Layout control results (VPEval Skill-based prompts).** Table 1 (left) displays the VPE-val accuracy on the VPEval Skill-based prompts. Our VIDEODIRECTORGPT significantly outperforms ModelScopeT2V and recent T2V/I2V generation models including AnimateD-iff (Guo et al., 2023), I2VGen-XL (Zhang et al., 2023), and SVD (Blattmann et al., 2023a) on all layout control skills. These results suggest that layouts generated by our LLM are highly accurate and greatly improve the control of object placements during video generation. Fig. 4 (left) shows an example where our LLM-generated *video plan* successfully guides Layout2Vid to accurately place the objects, while ModelScopeT2V fails to generate a 'pizza'. In Appendix J, we show additional examples (see Fig. 12) that our *video plan* can generate layouts requiring understanding of physics (*e.g.*, gravity, perspectives).

| Method | VPEval Skill-based | | | | | ActionBench-Direction |
|---|---|---|---|---|---|---|
| | Object | Count | Spatial | Scale | Overall Acc. (%) | Movement Direction Acc. (%) |
| ModelScopeT2V | 89.8 | 38.8 | 18.0 | 15.8 | 40.8 | 30.5 |
| AnimateDiff | 96.7 | 52.7 | 22.5 | 15.5 | 46.8 | 29.0 |
| I2VGen-XL | 96.5 | 62.0 | 35.2 | 23.7 | 54.3 | 35.2 |
| SVD | 93.1 | 46.7 | 29.2 | 15.0 | 45.7 | 30.7 |
| VIDEODIRECTORGPT | **97.1** | **77.4** | **61.1** | **47.0** | **70.6** | **46.5** |

Table 1: Evaluation on **VPEval Skill-based** and **ActionBench-Direction** prompts.

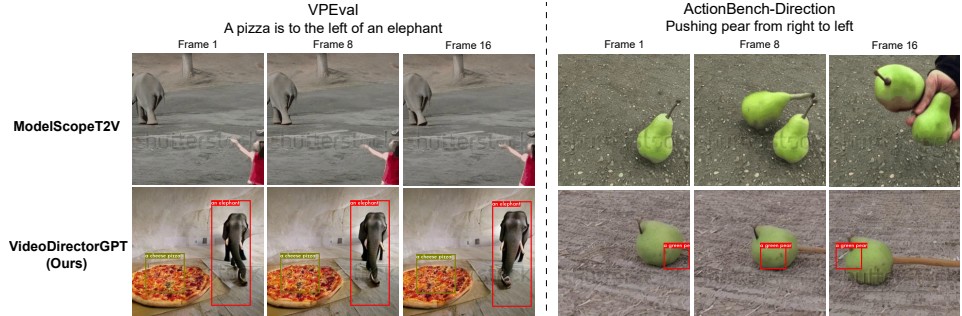

Figure 4: Generated examples on **VPEval Skill-based** and **ActionBench-Direction** prompts.

**Object movement results (ActionBench-Direction).** Table 1 (right) shows the performance on the ActionBench-Direction prompts. For each object, we give prompts that include four movement directions: 'left to right', 'right to left', 'top to bottom', and 'bottom to top'. Therefore, for a random guess, the movement accuracy is 25%. Our VIDEODIRECTORGPT outperforms ModelScopeT2V and other T2V/I2V models in object movement direction accuracy by a large margin, demonstrating that our LLM-generated layouts can improve the accuracy of object dynamics in video generation. Fig. 4 (right) shows that our LLM-generated *video plan* guides the Layout2Vid module to correctly locate and move the 'pear' in the accurate direction, whereas the 'pear' in the ModelScopeT2V video just gets grabbed.

**Open-domain results.** Table 2 presents the FVD, FID, and CLIPSIM scores on MSR-VTT. It's worth noting that our model is built on the pre-trained ModelScopeT2V backbone. Despite this, our method demonstrates good improvements in FVD and competitive scores in FID and CLIP-SIM. In addition, our method achieves better or comparable performance to models trained with larger video data (e.g., Make-A-Video) or with higher resolution (e.g., VideoLDM). Evaluation results on UCF-101 are presented in Appendix I, where we achieve significant improvement in FVD and similar Inception Score.

| Method | MSR-VTT | | |
|---|---|---|---|
| | FVD (↓) | FID (↓) | CLIPSIM (↑) |
| *Different arch / Training data* | | | |
| NUWA | — | 47.68 | 0.2439 |
| CogVideo (Chinese) | — | 24.78 | 0.2614 |
| CogVideo (English) | 1294 | 23.59 | 0.2631 |
| MagicVideo | 1290 | — | — |
| VideoLDM | — | — | 0.2929 |
| Make-A-Video | — | 13.17 | 0.3049 |
| *Same video backbone & Test prompts* | | | |
| ModelScopeT2V[+] | 683 | 12.32 | **0.2909** |
| VIDEODIRECTORGPT | **550** | **12.22** | 0.2860 |

Table 2: Comparison results on MSR-VTT. ModelScopeT2V[+]: Our replication on the same 2990 random test prompts.

| Method | ActivityNet Captions | | | Coref-SV | HiREST | |
| --- | --- | --- | --- | --- | --- | --- |
| | FVD ($\downarrow$) | FID ($\downarrow$) | Consistency ($\uparrow$) | Consistency ($\uparrow$) | FVD ($\downarrow$) | FID ($\downarrow$) |
| ModelScopeT2V | 980 | 18.12 | 46.0 | 16.3 | 1322 | 23.79 |
| ModelScopeT2V (with GT co-reference; oracle) | - | - | - | 37.9 | - | - |
| ModelScopeT2V (with prompts from our *video plan*) | - | - | - | - | 918 | 18.96 |
| VIDEODIRECTORGPT (Ours) | **805** | **16.50** | **64.8** | **42.8** | **733** | **18.54** |

Table 3: Multi-scene video generation on ActivityNet Captions, Coref-SV, and HiREST. *GT co-reference*: replacing co-reference pronouns in Coref-SV with the original object names (e.g., "his friends" may become "dog's friends"). *Prompts from our video plan*: an enhanced benchmark for ModelScopeT2V using scene descriptions from our *video plan*.

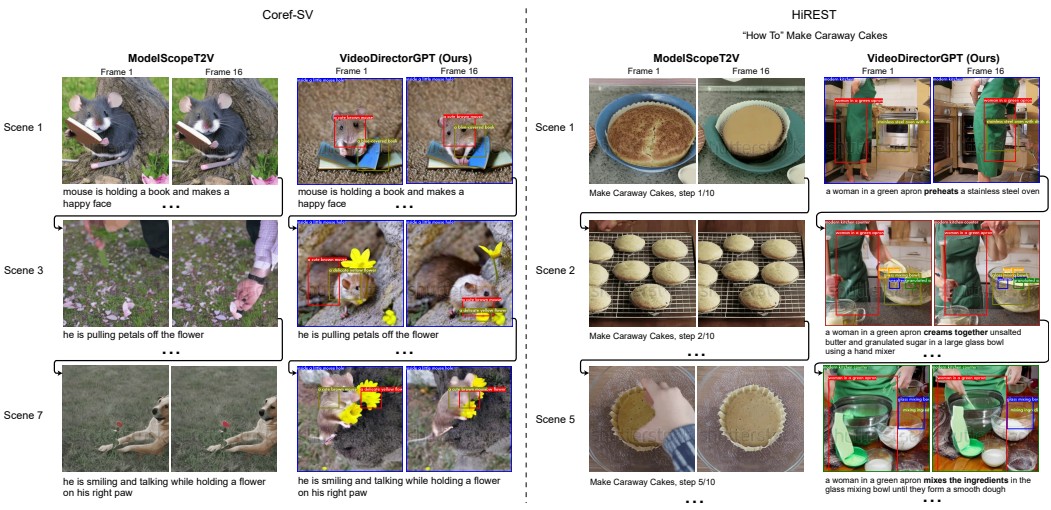

Figure 5: Generation examples on **Coref-SV (left)** and **HiREST (right)**.

## 5.2 Multi-Scene Video Generation

**Multiple sentences to multi-scene videos generation (ActivityNet Captions / Coref-SV).** As shown in the left two blocks of Table 3, our VIDEODIRECTORGPT outperforms ModelScopeT2V in visual quality (FVD/FID) and multi-scene object temporal consistency. Notably, for Coref-SV, our VIDEODIRECTORGPT achieves higher object consistency than ModelScopeT2V even with GT co-reference (where pronouns are replaced with their original noun counterparts, acting as oracle information; e.g., "she picked up ..." becomes "cat picked up ..."). As we do not have ground-truth videos for Coref-SV, we only use it to evaluate consistency when co-references are used across scenes. Fig. 5 (left) shows a video generation example from Coref-SV, where the LLM-generated *video plan* can guide the Layout2Vid module to generate the same mouse across scenes consistently, whereas ModelScopeT2V generates a hand and a dog instead of a mouse in later scenes.

**Single sentences to multi-scene videos generation (HiREST).** The right block of Table 3 shows our VIDEODIRECTORGPT achieves better visual quality scores (FVD/FID) than ModelScopeT2V on the HiREST dataset. Moreover, by comparing with an enhanced ModelScopeT2V benchmark, where videos are generated from each scene description in our GPT-4-produced video plan, we demonstrate the effectiveness of prompts generated by GPT-4. However, it still performs worse than our VIDEODIRECTORGPT, also showing the effectiveness of our Layout2Vid. As shown in Fig. 5 (right), our LLM can generate a step-by-step *video plan* from a single prompt and our Layout2Vid can generate consistent videos following the plan, showing how to make caraway cakes (a British seed cake). In contrast, ModelScopeT2V repeatedly generates the (visually inconsistent) caraway cake.

## 5.3 Additional Analysis

**Generating videos with custom image exemplars.** Our Layout2Vid can obtain CLIP image embeddings either from user-provided image exemplars or from entity text descriptions via

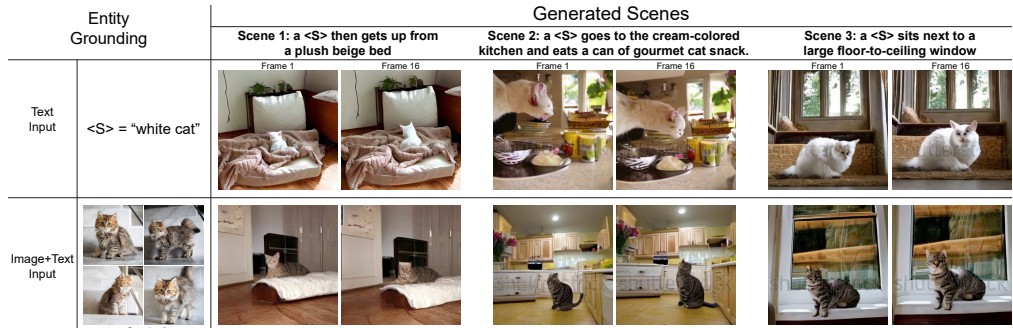

Figure 6: Video generation with text-only (top) and image+text (bottom) inputs with VIDEODIRECTORGPT. By giving text or image+text inputs, users can generate images where the identities of the entities are preserved across multiple scenes.

| # Denoising steps with layout guidance (out of 50) | MSR-VTT | | | ActionBench-Direction |
|---|---|---|---|---|
| | FVD (↓) | FID (↓) | CLIPSIM (↑) | Move Direction Acc. (%) |
| $\alpha = 0.1$ (5 steps) | 550 | **12.22** | **0.2860** | 46.5 |
| $\alpha = 0.2$ (10 steps) | 588 | 17.25 | 0.2700 | **59.8** |
| $\alpha = 0.3$ (15 steps) | 593 | 17.17 | 0.2702 | 57.8 |
| LLM-Dynamic-$\alpha$ (5-15 steps) | **523** | 13.75 | 0.2790 | 56.8 |

Table 4: Ablation of the denoising steps with layout guidance (via guided 2D attentions) in MSR-VTT and ActionBench-Direction prompts, where $\alpha = \frac{\text{\# steps with layout guidance}}{\text{\# total steps}}$.

the unCLIP Prior. In Fig. 6, we demonstrate that our Layout2Vid can flexibly take either text-only or image+text descriptions as input to generate consistent multi-scene videos.

**Dynamic layout strength control with GPT-4.** The number of denoising steps with layout guidance $\alpha$ controls layout control strength of Layout2Vid (see Appendix J for more detailed illustration). Instead of using a static $\alpha$ value, we explore dynamically adjusting it during the *video plan* generation by asking LLM how much layout guidance needs to be used for each prompt. Table 4 compares static $\alpha$ values and dynamic $\alpha$ values (LLM-Dynamic-$\alpha$), showing that LLMs help achieve a good balance in the quality-layout trade-off.

**Video Planner LLMs: GPT-4, GPT-3.5-turbo, and LLaMA2.** We conducted an ablation study using the open-sourced LLaMA2-7B and LLaMA2-13B models (Touvron et al., 2023b) as well as GPT-3.5-turbo in the video planning stage on MSR-VTT and ActionBench-Directions. We evaluated LLaMA2 in both fine-tuned and zero-shot settings. To collect data for fine-tuning LLaMA2, we randomly sampled 2000 prompts from WebVid-10M (Bain et al., 2021) and generated layouts with GPT-4. In addition to the visual quality (FID/FVD) and video-text alignment (CLIPSIM) scores on MSR-VTT and movement direction accuracy score on ActionBench-Directions, we also report the number of samples whose output layouts can be successfully parsed (*e.g.*, #Samples). Evaluation setup details are given in Appendix H. Table 5 shows that zero-shot LLaMA2-13B struggles to generate layouts that can be successfully parsed (*e.g.*, only 82 out of 600). It also achieves worse scores on all three metrics (FID, FVD, and CLIPSIM) on MSR-VTT, as well as the movement direction accuracy on ActionBench-Directions, which shows that GPT models have stronger in-context learning skills and can generate more reasonable layouts. On the other hand, all scores can be improved by fine-tuning on layouts generated by GPT-4. Fine-tuning open-sourced LLMs can be a future direction worthwhile to explore. Another point to note is that GPT-3.5-turbo and GPT-4 achieve very similar performance on MSR-VTT, but GPT-3.5-turbo performs significantly worse than GPT-4 on ActionBench-Directions. This suggests that GPT-3.5-turbo is less effective at handling prompts requiring strong layout control compared to GPT-4.

**Challenging cases.** The bounding box-based control of our method poses two main challenges. Firstly, some objects might not follow the bounding box control well when there are too many overlapping bounding boxes. For example, in Fig. 5 right, the "hand mixer" and

| LLMs | MSR-VTT | | | | ActionBench-Directions | |
|------|---------|---------|-----------|------------|---------------------------|---------------|
|      | FVD ($\downarrow$) | FID ($\downarrow$) | CLIPSIM ($\uparrow$) | #Samples ($\uparrow$) | Movement Direction Acc (%) | #Samples ($\uparrow$) |
| *Fine-tuned* | | | | | | |
| LLaMA2-7B | 580 | 12.69 | 0.2851 | 2749 | 35.8 | 537 |
| LLaMA2-13B | 556 | 12.40 | 0.2854 | 2786 | 53.3 | 561 |
| *Zero-shot* | | | | | | |
| LLaMA2-13B | 573 | 13.47 | 0.2792 | 2236 | - | 82 |
| GPT-3.5-turbo | 558 | 12.27 | 0.2852 | **2990** | 49.0 | **600** |
| GPT-4 (default) | **550** | **12.22** | **0.2860** | **2990** | **59.8** | **600** |

Table 5: Ablation of video generation with video plans generated from different LLMs on MSR-VTT and ActionBench-Directions.

"unsalted butter", which have relatively small bounding box sizes, do not follow the box control well. Secondly, the control of whether the background is static or moving is usually determined by the prior knowledge of the video generation model. For example, as shown in Fig. 13, it's hard for Layout2Vid to generate a moving bottle with a static background and a moving boat with a completely static background (see paragraph "Object movement prompts with different types of objects" in Appendix J.2 for details).

## 5.4 Human Evaluation

**Human preference.** We conduct a human evaluation study on the multi-scene videos generated by both our VIDEODIRECTORGPT and ModelScopeT2V on the Coref-SV dataset. We show 50 videos to 10 crowd-annotators from AMT[1] to rate each video (56 annotators in total, 10 annotators per

| Evaluation category | Human Preference (%) $\uparrow$ | | |
|---------------------|------|--------------|-----|
|                     | Ours | ModelScopeT2V | Tie |
| Quality | **50** | 34 | 16 |
| Text-Video Alignment | **58** | 36 | 6 |
| Object Consistency | **62** | 28 | 10 |

Table 6: Human preference on multi-scene videos generated with Coref-SV prompts.

video) and calculate human preferences for each video with average ratings. Table 6 shows that the multi-scene videos (with Coref-SV prompts) generated by our VIDEODIRECTORGPT are preferred by human annotators than ModelScopeT2V videos in all three categories (Quality, Text-Video Alignment, and Object Consistency).

**Step-by-step error analysis.** In our error analysis of video planning and generation stages, we measure 1-5 Likert scale accuracy of four intermediate generation steps: scene descriptions, layouts, consistency groupings, and final video. While the first three planning steps achieve high scores ($\geq$ 4.52), there is a big score drop in the layout-guided video generation (4.52 $\rightarrow$ 3.61). This suggests that our VIDEODIRECTORGPT could generate more accurate videos, once we have access to a stronger T2V backbone than ModelScopeT2V. See Appendix G for more detailed analysis.

## 6 Conclusion

In this work, we propose VIDEODIRECTORGPT, a novel framework for consistent multi-scene video generation, leveraging the knowledge of LLMs for video content planning and grounded video generation. In the first stage, we employ GPT-4 as a video planner to craft a *video plan*, a multi-component script for videos with multiple scenes. In the second stage, we use Layout2Vid, a grounded video generation module, to generate videos with layout and consistency control. Experiments demonstrate that our proposed framework substantially improves object layout and movement control over state-of-the-art methods on open-domain single-scene video generation, and can generate consistent multi-scene videos while maintaining visual quality.

---

[1]Amazon Mechanical Turk: https://www.mturk.com

## Acknowledgments

We thank the reviewers for the thoughtful discussion and feedback. This work was supported by ARO W911NF2110220, DARPA MCS N66001-19-2-4031, DARPA KAIROS Grant FA8750-19-2-1004, NSF-AI Engage Institute DRL211263, ONR N00014-23-1-2356, Accelerate Foundation Models Research program, and a Bloomberg Data Science Ph.D. Fellowship. The views, opinions, and/or findings contained in this article are those of the authors and not of the funding agency.

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

# Appendix

# A    Reproducibility Statement

Our model is built upon the publicly available code repository from GLIGEN (Li et al., 2023)[2] and ModelScopeT2V (Wang et al., 2023b)[3]. Please see Sec. 3.2/Appendix D for model architecture details, Sec. 4/Appendix F.2/Appendix F.3 for dataset details. **We provide code and video samples (`video_samples.html`) in the supplementary material.**

# B    Additional Related Works

**Text-to-video generation.**    The text-to-video (T2V) generation task is to generate videos from text descriptions. Early T2V generation models (Li et al., 2017; 2019b) used variational autoencoders (VAE) (Kingma & Welling, 2014) and generative adversarial networks (GAN) (Goodfellow et al., 2020), while multimodal language models (Hong et al., 2022; Wu et al., 2022a; Villegas et al., 2023; Maharana et al., 2022; Ge et al., 2022; Wu et al., 2021) and denoising diffusion models (Ho et al., 2022; Singer et al., 2023; Blattmann et al., 2023b; Khachatryan et al., 2023; Wang et al., 2023a; Yin et al., 2023) have become popular for recent works. Since training a T2V generation model from scratch is computationally expensive, recent work often leverages pre-trained text-to-image (T2I) generation models such as Stable Diffusion (Rombach et al., 2022) by finetuning them on text-video pairs (Wang et al., 2023b; Blattmann et al., 2023b). While this warm-start strategy enables high-resolution video generation, it comes with the limitation of only being able to generate short video clips, as T2I models lack the ability to maintain consistency through long videos. Recent works on long video generation (Blattmann et al., 2023b; Yin et al., 2023; Villegas et al., 2023; He et al., 2023) aim at generating long videos of a few minutes. However, the generated videos often display the continuation or repetitive patterns of a single action (*e.g.*, driving a car) instead of transitions and dynamics of multiple changing actions/events (*e.g.*, five steps about how to bake a cake). In this work, we address this problem of multi-scene video generation with a two-stage framework: using an LLM (*e.g.*, GPT-4) to generate a structured *video plan* (consisting of stepwise scene descriptions, entities and their layouts) and generating videos using Layout2Vid, a layout-guided text-to-video generation model with consistency control. Our Layout2Vid infuses layout control and multi-scene temporal consistency into a pretrained T2V generation model via data and parameter-efficient training, while preserving its original visual quality.

# C    Video Planning

## C.1    GPT-4 Prompt Templates

In this section, we provide the prompt templates we give to our video planner (Sec. 3.2). The *video plan* is generated in two steps by prompting GPT-4[4] with different in-context examples (we use 1 and 5 in-context examples for the first and second steps, respectively). In the first step (see Fig. 7), we ask GPT-4 to expand a single text prompt into a multi-scene *video plan*. Each scene comes with a text description, a list of entities, and a background. In the second step (see Fig. 8), we generate a list of bounding boxes for the entities in each frame based on the list of entities and the scene description. In line with VPGen (Cho et al., 2023b), we utilize the $[x_0, y_0, x_1, y_1]$ format for bounding boxes, where each coordinate is normalized to fall within the range [0,1]. For in-context examples, we present 0.05 as the minimum unit for the bounding box, equivalent to a 20-bin quantization over the [0,1] range.

---

[2]GLIGEN: https://github.com/gligen/GLIGEN

[3]ModelScopeT2V: https://modelscope.cn/models/damo/text-to-video-synthesis/summary, https://github.com/ExponentialML/Text-To-Video-Finetuning/tree/main

[4]We employ the `gpt-4-0613` version.

**Instructions:**
Given a single text prompt, you need to envision a multi-scene video by generating a sequence of stepwise prompts to describe the text prompt. For each step, you also need to generate the set of entities needed and describe the background scene where the video should occur. Related steps should maintain similar entities and background scenes. Before you write each stepwise description, you must follow these instructions:
1. Each step prompt must contain only a single motion or action.
2. Each step prompt must include all relevant objects and describe the environment scene.
3. Make sure each step prompt must be easy described by a shot video cilp of 8 seconds at 2fps.
Use the following format: [[step_1: [prompt, entities, background]; step_2: [prompt, entities, background]; … step_n: [prompt, entities, background]]

**In-context example:**
**Input:** Provide stepwise prompts for the text prompt: make butter biscuits
**Output:**
Step 1: [Prompt: A chief preheats the oven; Entities: chief, oven; Background: kitchen counter];
Step 2: [Prompt: A chief creams together butter and sugar in a mixing bowl; Entities: chief, mixing bowl, butter, sugar, hand mixer; Background: kitchen]
Step 3: [Prompt: A chief beats in eggs into the mixing bowl; Entities: chief, eggs, mixing bowl; Background: kitchen];
  …

**Input text:** Provide stepwise prompts for the text prompt: [*insert single text prompt here*]

Figure 7: Prompt template for the 1st video planning step (scene descriptions and entities/background generation).

**Instructions:**
Assuming the frame size is normalized to the range 0-1, you need to give a possible 8-frame layout with bounding boxes of the listed entities of a given scene description. Each object in the image is one rectangle or square box in the layout and size of boxes should be as large as possible. You need to generate layouts from the close up camera view of the event. The layout difference between two adjacent frames must be small, considering the small interval. You also need to generate a caption that best describes the image for each frame. After generating all frames, add reasoning to your design.
Use format:
Frame_1: [[object1, [left, top, right, bottom]], [object2, [left, top, right, bottom]], ..., [object_n, [left, top, right, bottom]]], caption:...
Frame_2: [[object1, [left, top, right, bottom]], [object2, [left, top, right, bottom]], ..., [object_n, [left, top, right, bottom]]], caption:...
…
Frame_8: [[object1, [left, top, right, bottom]], [object2, [left, top, right, bottom]], ..., [object_n, [left, top, right, bottom]]], caption:...
Reasoning:...

**In-context example:**
**Input:** Please generate bounding box coordinates for the following entities based on the scene description.
Entities: ['table', 'cat', 'chair']. Scene description: A cat jumps from a chair to a table.
**Output:**
Frame_1: [["table", [0.1, 0.6, 0.9, 1.0]], ["cat", [0.4, 0.55, 0.6, 0.6]], ["chair", [0.05, 0.6, 0.4, 1.0]]], caption: A cat is standing on the table, preparing to jump.
Frame_2: [["table", [0.1, 0.6, 0.9, 1.0]], ["cat", [0.45, 0.5, 0.65, 0.55]], ["chair", [0.05, 0.6, 0.4, 1.0]]], caption: The cat has just left the table and is in mid-air.
…
Reasoning: The cat\'s bounding box in Frame 1 is on the table, showing the cat in the take-off position. From Frame 2 to Frame 6, the cat is shown in various stages mid-flight, with the cat\'s bounding box moving closer to the chair with each frame. …

**Input text:** Provide bounding box coordinates for the prompt: [*insert entities and scene description here*]

Figure 8: Prompt template for the 2nd video planning step (layout generation).

## C.2 API Cost

Using GPT-4 tokenizer, the average input/output token lengths of each step are 2K/1K for the first step and 6K/1K for the second step. Using GPT-4, it takes 0.12 USD and 0.24 USD for the inference of the first and second steps, respectively.

# D   Layout2Vid

## D.1   Preliminaries: T2V Generation Backbone

We implement Layout2Vid by injecting layout control capability into ModelScopeT2V (Wang et al., 2023b), a public text-to-video generation model based on Stable Diffusion (Rombach et al., 2022). ModelScopeT2V consists of (1) a CLIP ViT-H/14 (Radford et al., 2021) text encoder, (2) an autoencoder, and (3) a diffusion UNet (Ronneberger et al., 2015; Ho et al., 2020). Given a $T$ frame video $x \in \mathbb{R}^{T \times 3 \times H \times W}$ with video caption $c$ and frame-wise layouts $\{e\}_{i=1}^{T}$, ModelScopeT2V first uses an autoencoder to encode the video into a latent representation $z = \mathcal{E}(x)$. The diffusion UNet performs denoising steps in the latent space to generate videos, conditioned on the CLIP text encoder representation of video captions. The UNet comprises a series of spatio-temporal blocks, each containing four modules: spatial convolution, temporal convolution, spatial attention, and temporal attention. Since the original ModelScopeT2V does not offer control beyond the text input, we build upon the 2D

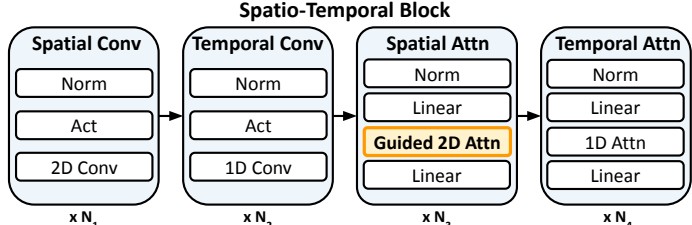

Figure 9: **Spatio-temporal blocks within the diffusion UNet of Layout2Vid**. The spatio-temporal block comprises four modules: spatial convolution, temporal convolution, spatial attention, and temporal attention. We adopt settings from ModelScopeT2V, where (N1, N2, N3, N4) are set to (2, 4, 2, 2). We use "Act" to represent activation layers.

attention module in the spatial attention module to create 'Guided 2D Attention', which allows for spatial control using bounding boxes.

### D.2    Layout2Vid Spatio-Temporal Blocks

Fig. 9 illustrates the spatio-temporal blocks within the diffusion UNet of Layout2Vid. The spatio-temporal block comprises four modules: spatial convolution, temporal convolution, spatial attention, and temporal attention. Following Li et al. (2023), we build upon the 2D attention to create the guided 2D attention, which enables layout-guided video generation with explicit spatial control over a list of entities represented by their bounding boxes, as well as visual and text content.

### D.3    Training and Inference Details

Table 7 contains the model architecture details and training/inference parameter settings for our Layout2Vid.

**Training.**    The highlight of our Layout2Vid training is that it was conducted solely on image-level data with bounding box annotations. We trained the MLP layers for grounding tokens and the gated self-attention layers with the same bounding-box annotated data used in GLIGEN (Li et al., 2023), which consists of 0.64M images. We train Layout2Vid for 50k steps, which takes only two days with 8 A6000 GPUs (each 48GB memory). All the remaining modules in the spatio-temporal block (see main paper Fig. 4) are frozen during training. We illustrate the training and inference procedure of Layout2Vid in main paper Fig. 3.

**Inference.**    During inference, we use the Karlo implementation of unCLIP Prior to the entities to convert the texts into their corresponding CLIP image embeddings, and CLIP text encoder to get their corresponding text embeddings. We use CLIP ViT-L/14 as a backbone during training to be consistent with Karlo. This helps us to preserve the visual consistency of the same object by using the same image embedding across scenes. In addition, we use PLMS Rombach et al. (2021) (based on PNDM Liu et al. (2022)) as our default sampler following GLIGEN (Li et al., 2023), since we found no consistent improvements for visual quality and video-text alignment scores when switching to the DDIM (Song et al., 2020) sampler used in ModelScopeT2V. The comparison of the PLMS and DDIM samplers is shown in Table 8.

## E    Human-in-the-Loop Video Plan Editing

One useful property of our VIDEODIRECTORGPT is that it is very flexible to edit *video plan*s via human-in-the-loop editing. Given an LLM-generated *video plan*, users can generate customized content by adding/deleting/replacing the entities, adding/changing the background, and modifying the bounding box layouts of entities. Figure 10 showcases the

| Architecture | | Training | |
|---|---|---|---|
| $z$-shape | $32 \times 32 \times 4$ | # Train steps | 50k |
| Channels | 320 | Learning rate | 5e-5 |
| Depth | 2 | Batch size per GPU | 32 |
| Attention scales | 1,0.5,0.25 | # GPUs | 8 |
| Number of heads | 8 | GPU-type | A6000 (48GB) |
| Head dim | 64 | Loss type | MSE |
| Channel multiplier | 1,2,4,4 | Optimizer | AdamW |
| **CA Conditioning** | | **Inference** | |
| Context dimension | 1024 | Sampler | PLMS |
| Sequence length | 77 | # denoising steps | 50 |

Table 7: Hyperparameters for Layout2Vid.

| Samplers | FVD ($\downarrow$) | FID ($\downarrow$) | CLIPSIM ($\uparrow$) |
|---|---|---|---|
| DDIM | **508** | 12.52 | 0.2848 |
| PLMS | 550 | **12.22** | **0.2860** |

Table 8: Comparison between PLMS and DDIM sampler on MSR-VTT.

example videos generated from the text prompt "A horse running", and videos generated from the *video plan* modified by users (*i.e.*, changing the object sizes and backgrounds).

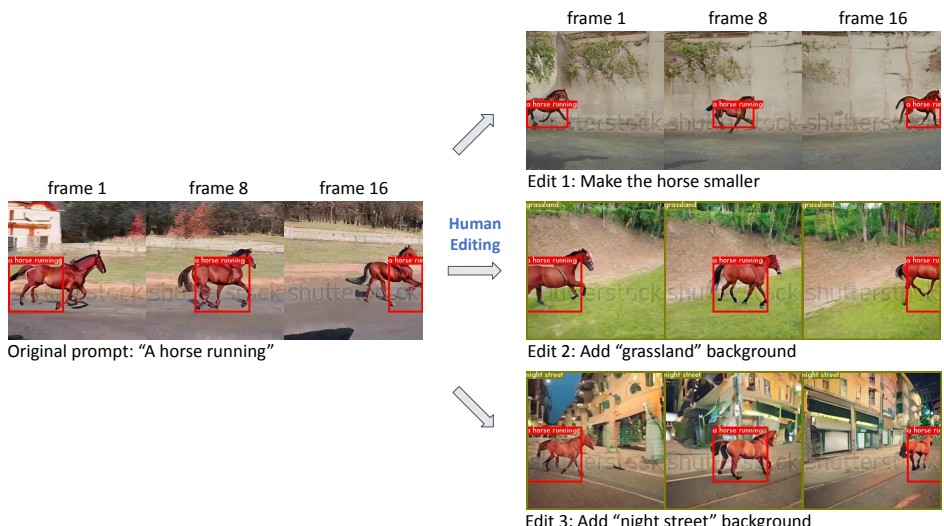

Figure 10: Video generation examples for human-in-the-loop editing. Users can modify the video plan (*e.g.*, add/delete objects, change the background and entity layouts, *etc.*) to generate customized video contents. Given the same text prompt "A horse running", we provide visualizations with a smaller horse and different backgrounds (*i.e.*, "night street" and "grassland").

## F  Experiment Setup

We provide additional details on our experiment setups (Sec. 4 in the main paper) below.

### F.1  Evaluated Models

We compare our VIDEODIRECTORGPT to 6 popular T2V generation models, NUWA (Wu et al., 2022b), CogVideo (Hong et al., 2022), VideoLDM (Blattmann et al., 2023b), MagicVideo (Zhou et al., 2022), Make-A-Video (Singer et al., 2023), and ModelScopeT2V (Wang

et al., 2023b). Since NUWA, VideoLDM, MagicVideo, Make-A-Video, and CogVideo (English) are not publicly available, we primarily compare our VIDEODIRECTORGPT with ModelScopeT2V, and present comparisons with the other models on the datasets for which their papers have provided results. ModelScopeT2V is the closest baseline to our framework among all these models, because our Layout2Vid utilizes its frozen weights and only trains a small set of new parameters to add spatial control and temporal consistency across multiple scenes.

### F.2 Prompts for Single-Scene Video Generation

For single-scene video generation, we conduct experiments with VPEval Skill-based prompts to evaluate layout control (Cho et al., 2023b), ActionBench-Direction prompts to assess object dynamics (Wang et al., 2023c), and MSR-VTT to cover diverse open-domain scenes (Xu et al., 2016).

**VPEval Skill-based prompts** evaluate different object-centric layout control skills in text-to-image/video generation. We randomly sample 100 prompts for each of the four skills: Object (generation of a single object), Count (generation of a specific number of objects), Spatial (generation of two objects with a spatial relation; *e.g.*, left/right/above/below), and Scale (generation of two objects with a relative scale relation; *e.g.*, bigger/smaller/same).

**ActionBench-Direction prompts** evaluate the action dynamics (object movement directions) in video language models. We prepare the prompts by sampling video captions from ActionBench-SSV2 (Wang et al., 2023c) and balancing the distribution of movement directions. Concretely, we select captions from the ActionBench-SSV2 validation split that include phrases like 'right to left' or 'left to right' (*e.g.*, 'pushing a glass from left to right'), which are common phrases describing movement directions in the captions. Then we augment these prompts by switching the directions to each of four directions: 'left to right', 'right to left', 'top to bottom', and 'bottom to top' to create 100 prompts for each direction. We call the resulting 400 prompts as *ActionBench-Direction* prompts. These prompts ensure a balanced distribution of movement directions while maintaining diversity in objects.

**MSR-VTT** is an open-domain video captioning dataset, which allows us to check if our Layout2Vid maintains the original visual quality and text-video alignment performance of the ModelScopeT2V backbone after integration of the layout/movement control capabilities. The MSR-VTT test set comprises 2,990 videos, each paired with 20 captions. Following VideoLDM (Blattmann et al., 2023b), we sample one caption from the 20 available captions for each video and use the 2,990 corresponding generated videos for evaluation.

### F.3 Prompts for Multi-Scene Video Generation

For multi-scene video generation, we experiment with two types of input prompts: (1) a list of sentences describing events – ActivityNet Captions (Krishna et al., 2017) and Coref-SV prompts based on Pororo-SV (Li et al., 2019b) and (2) a single sentence from which models generate multi-scene videos – HiREST (Zala et al., 2023).

**ActivityNet Captions** is a dense-captioning dataset designed for detecting and describing multiple events in videos using natural language. For the multi-scene video generation task, we use 165 randomly sampled videos from the validation split and use the event captions as input for ModelScopeT2V and our VIDEODIRECTORGPT. When calculating object consistency, we find the subject of the first event caption via spaCy dependency parser (Honnibal & Montani, 2017) and check its appearance in multiple scenes.

**Coref-SV** is a new multi-scene text description dataset that we propose to evaluate the consistency of object appearances across multi-scene videos. We prepare the Coref-SV prompts by augmenting the Pororo-SV dataset (Li et al., 2019b; Kim et al., 2017), which consists of multi-scene paragraphs from the "Pororo the Little Penguin" animated series. To evaluate the temporal consistency of video generation models trained on real-world videos, we extend its original animation characters (*e.g.*, Pororo) to humans and common animals and examine their appearance across different scenes. Concretely, we sample 10 episodes, each consisting of multiple scenes (6.2 scenes per episode on average). Then,

we replace the first appearance of a character with one of the predefined 10 real-world entities (*e.g.*, person/dog, etc.) and replace the remaining appearances of the character with pronouns (*e.g.*, he/she/it/etc.). In total, we obtain 100 episodes (=10 episodes × 10 entities) in Coref-SV. In order to generate visually consistent entities, the multi-scene video generation models would need to address the co-reference of these target entities across different scenes. We use the final scene descriptions as input for both ModelScopeT2V and VIDEODIRECTORGPT. When calculating object consistency, we use the selected entity as the target object.

**HiREST** provides step annotations for instructional videos paired with diverse 'How to' prompts (*e.g.*, a video paired with 'how to make butter biscuits' prompt is broken down into a sequence of short video clips of consecutive step-by-step instructions). For the multi-scene video generation task, we employ 175 prompts from the test splits, where we only include the prompts with step annotations, to ensure that it is possible to create multi-scene videos from the prompts. Note that instead of providing a list of scene description sentences like in ActivityNet Captions/Coref-SV, we only give the single high-level 'How to' prompt and let the models generate a multi-scene video from it. In VIDEODIRECTORGPT, our LLM can automatically generate the multi-scene *video plan* and video from the input prompt. In contrast, for the ModelScopeT2V baseline, we help the model understand the different number of scenes to generate by pre-defining the number of scenes $N$, and independently generate $N$ videos by appending the suffix "step $n/N$" to the prompt for $n$-th scene (*e.g.*, "Cook Beet Greens, step 1/10"). To ensure that our VIDEODIRECTORGPT videos and ModelScopeT2V videos are equal in length, we use the same number of scenes generated by our LLM during the planning stage for ModelScopeT2V. As mentioned in main paper Sec. 5.2, we also try an ablation with giving ModelScopeT2V the LLM generated prompts instead of just "step $n/N$".

## G  Human Evaluation Details

We provide details of our human evaluation and error analysis setups, as well as the error analysis results described in Sec. 4.

**Human evaluation details.**  We conduct a human evaluation study on the multi-scene videos generated by both our VIDEODIRECTORGPT and ModelScopeT2V on the Coref-SV dataset. Since we know the target entity and its co-reference pronouns in the Coref-SV prompts, we can compare the temporal consistency of the target entities across scenes. We evaluate the human preference between videos from two models in each category of Quality, Text-Video Alignment, and Object Consistency:

- *Quality:* it measures how well the video looks visually.
- *Text-Video Alignment:* it assesses how accurately the video adheres to the input sentences.
- *Object Consistency:* it evaluates how well the target object maintains its visual consistency across scenes.

We show 50 videos to ten crowd-annotators from AMT[5] to rate each video (28 unique annotators, ten annotators rate each video) and calculate human preferences for each video with average ratings. All videos are randomly shuffled such that annotators do not know which model generated each video. To ensure high-quality annotations, we require they have an AMT Masters, have completed over 1000 HITs, have a greater than 95% approval rating, and are from one of the United States, Great Britain, Australia, or Canada (as our task is written in the English language). We pay annotators $0.10 to evaluate a video (roughly $12-14/hr).

**Step-by-step error analysis details.**  We conduct an error analysis on each step of our single sentence to multi-scene video generation pipeline for HiREST prompts. We analyze

---

[5]Amazon Mechanical Turk: https://www.mturk.com

the generated multi-scene text descriptions, layouts, and entity/background consistency groupings to evaluate our video planning stage and the final video to evaluate the video generation stage.

- *Multi-Scene Text Descriptions Accuracy*: we measure how well these descriptions depict the intended scene from the original prompt (*e.g.*, if the original prompt is "Make buttermilk biscuits" the descriptions should describe the biscuit-making process and not the process for pancakes).
- *Layout Accuracy*: we measure how well the generated layouts showcase a scene for the given multi-scene text description (*e.g.*, the bounding boxes of ingredients should go into a bowl, pan, etc. instead of randomly moving across the scene).
- *Entity/Background Consistency Groupings Accuracy*: we measure how well the generated entities and backgrounds are grouped (*e.g.*, entities/backgrounds that should look consistent throughout the scenes should be grouped together).
- *Final Video Accuracy*: we measure how well the generated video for each scene matches the multi-scene text description (*e.g.*, if the multi-scene text description is "preheating an oven", the video should accurately reflect this).

We ask an expert annotator to rank the generations (multi-scene text description, layouts, etc.) on a Likert scale of 1-5 for 50 prompts/videos. Analyzing the errors at each step enables us to check which parts are reliable and which parts need improvement. As a single prompt/video can contain many scenes, to simplify the process for layout and final video evaluation of a prompt/video, we sub-sample three scene layouts and corresponding scene videos and average their scores to obtain the "Layout Accuracy" and "Final Video Accuracy."

**Step-by-step error analysis results.** As shown in Table 9, our LLM-guided planning scores high accuracy on all three components (up to 4.52), whereas the biggest score drop happens in the layout-guided video generation (4.52 → 3.61). This suggests that our VIDEODIRECTORGPT could generate even more accurate videos, once we have access to a stronger T2V backbone than ModelScopeT2V.

| Stage 1: Video Planning (with GPT-4) | | | Stage 2: Video Generation (with Layout2Vid) |
|---|---|---|---|
| Multi-scene Text Descriptions (↑) | Layouts (↑) | Entity/Background Consistency Groupings (↑) | Final Video (↑) |
| 4.92 | 4.69 | 4.52 | 3.61 |

Table 9: Step-wise error analysis of VIDEODIRECTORGPT video generation pipeline on HiREST prompts. We use a Likert scale (1-5) to rate the accuracy of the generated components at each step.

# H Ablation Studies

In this section, we provide ablation studies on our design choices, including using different LLMs for video planning, the number of layout-guided denoising steps, different embeddings for layout groundings, and layout representation formats.

## H.1 Video Planner LLMs: GPT-4, GPT-3.5-turbo, and LLaMA2

We conducted an ablation study using the open-sourced LLaMA2-7B and LLaMA2-13B models (Touvron et al., 2023b) as well as GPT-3.5-turbo in the video planning stage on MSR-VTT and ActionBench-Directions.

**Evaluation settings.** In this study, we evaluated LLaMA2 in both fine-tuned and zero-shot settings. For the zero-shot evaluation, we used the same number of in-context examples (*i.e.*, 5 examples) for both GPT-3.5-turbo and GPT-4, but only 3 in-context examples for the LLaMA2 models due to their 4096 token length limits. To collect data for fine-tuning

LLaMA2, we randomly sampled 2000 prompts from WebVid-10M (Bain et al., 2021) and generated layouts with GPT-4 using the same prompts as described in our main paper. We fine-tuned the LLaMA2 models for 1000 steps with a learning rate of 2e-4 and a cosine scheduler. In addition to the visual quality (FID/FVD) and video-text alignment (CLIPSIM) scores on MSR-VTT and movement direction accuracy score on ActionBench-Directions, we also report the number of samples whose output layouts can be successfully parsed (*e.g.*, #Samples). We consider the output valid if it contains 9 frames, and each frame includes at least one object with a bounding box layout. Videos are generated with $\alpha = 0.1$ for MSR-VTT and $\alpha = 0.2$ for ActionBench-Directions.

## H.2    Entity Grounding Embeddings: Image v.s. Text

As discussed in Sec. 3.2 in our main paper, we compare using different embeddings for entity grounding on 1000 randomly sampled MSR-VTT test prompts. As shown in Table 10, CLIP image embedding is more effective than CLIP text embedding, and using the CLIP image-text joint embedding yields the best results. Thus, we propose to use the image+text embeddings for the default configuration.

| Entity Grounding | MSR-VTT | | | Coref-SV |
|---|---|---|---|---|
| | FVD ($\downarrow$) | FID ($\downarrow$) | CLIPSIM ($\uparrow$) | Consistency (%) |
| Image Emb. | 737 | 18.38 | 0.2834 | 42.6 |
| Text Emb. | 875 | 23.18 | 0.2534 | 36.9 |
| Image+Text Emb. (default) | **606** | **14.60** | **0.2842** | **42.8** |

Table 10: Ablation of entity grounding embeddings of our Layout2Vid on MSR-VTT and Coref-SV.

## H.3    Layout Control: Bounding Box v.s. Center Point

In Table 11, we compare different layout representation formats on 1000 randomly sampled MSR-VTT test prompts. We use image embedding for entity grounding and $\alpha = 0.2$ for layout control. Compared with no layout ('w/o Layout input') or center point-based layouts (without object shape, size, or aspect ratio), the bounding box based layout guidance gives better visual quality (FVD/FID) and text-video alignment (CLIPSIM).

| Layout representation | FVD ($\downarrow$) | FID ($\downarrow$) | CLIPSIM ($\uparrow$) |
|---|---|---|---|
| w/o Layout input | 639 | 15.28 | 0.2777 |
| Center point | 816 | 18.65 | 0.2707 |
| Bounding box (default) | **606** | **14.60** | **0.2842** |

Table 11:  Ablation of layout representation of our VIDEODIRECTORGPT on MSR-VTT.

## H.4    Trainable Layers: Gated Self-Attention Only v.s. Entire Guided 2D Attention

**Training settings.**    In our Layout2Vid, only the MLP layers for grounding tokens and the gated self-attention layers are trained, accounting for 13% of the total parameters. We conducted an ablation study to unfreeze more parameters from the ModelScopeT2V backbone. Specifically, all layers in the guided 2D attention module, such as self-attention, gated self-attention, and cross-attention, are made trainable, which constitutes 27% of the total parameters. Starting from the model trained in our main paper, we further train this variant with more trainable parameters for 50k steps, keeping all other training hyper-parameters constant.

**Result analysis.**    As shown in Table 12, unfreezing all parameters in the guided 2D attention module results in significantly worse performance compared to training only the additional gated self-attention layers. This decline in performance occurs because the temporal attention and temporal convolution layers, which are inactive during our image-data

| Trainable Layers | FVD ($\downarrow$) | FID ($\downarrow$) | CLIPSIM ($\uparrow$) |
|---|---|---|---|
| Entire Guided 2D Attn (GatedSA + SelfAttn + CrossAttn) | 1626 | 70.47 | 0.2231 |
| GatedSA only (default) | **550** | **12.22** | **0.2860** |

Table 12: Comparison of only updating parameters in gated self-attention (GatedSA) between updating all parameters in guided 2D attention (Entire Guided 2D Attn) of our VIDEODIRECTORGPT on MSR-VTT.

training, cause misalignment in the spatial and temporal layers during inference when the temporal layers are activated.

# I Additional Experiments

## I.1 Comparison on UCF-101

In addition to the MSR-VTT dataset for open-domain single-scene video generation, we also conduct experiments for the UCF-101 dataset (Soomro et al., 2012). Since ModelScopeT2V does not report their evaluation results on UCF-101 dataset in their technical report, we use their publicly released checkpoint and run evaluation on the same 2048 randomly sampled test prompts as used in our VideoDirectorGPT evaluation. Compared with ModelScopeT2V, we achieve a significant improvement in FVD (ModelScopeT2V 1093 vs. Ours 748) and competitive performance in the Inception Score (ModelScopeT2V 19.49 vs. Ours 19.42). In addition, we notice that ModelScopeT2V is not good at video generation on this UCF-101 dataset, which is also observed in other recent works (He et al., 2023). We could expect our VideoDirectorGPT to perform better with stronger T2V backbones.

| Method | UCF-101 | |
|---|---|---|
| | FVD ($\downarrow$) | IS ($\uparrow$) |
| *Different arch / Training data* | | |
| LVDM | 917 | — |
| CogVideo (Chinese) | 751 | 23.55 |
| CogVideo (English) | 701 | 25.27 |
| MagicVideo | 699 | — |
| VideoLDM | 550 | 33.45 |
| Make-A-Video | 367 | 33.00 |
| *Same video backbone & Test prompts* | | |
| ModelScopeT2V[†] | 1093 | **19.49** |
| VIDEODIRECTORGPT | **748** | 19.42 |

Table 13: Evaluation results on UCF-101. ModelScopeT2V[†]: Our replication with 2048 randomly sampled test prompts.

# J Additional Visualization and Qualitative Examples

## J.1 Visualization of Different Layout-Guided Denoising Steps.

**Balancing layout control strength with visual quality.** During video generation, we use two-stage denoising in Layout2Vid following Li et al. (2023), where we first use layout-guidance with Guided 2D attention for $\alpha * N$ steps, and use the denoising steps without Guided 2D attention for the remaining $(1 - \alpha) * N$ steps, where $N$ is the total number of denoising steps, and $\alpha \in [0, 1]$ is the ratio of layout-guidance denoising steps. In our additional analysis (see main paper Table 4), we find that a high $\alpha$ generally increases layout control but could lead to lower visual quality, which is also consistent with the finding in Li

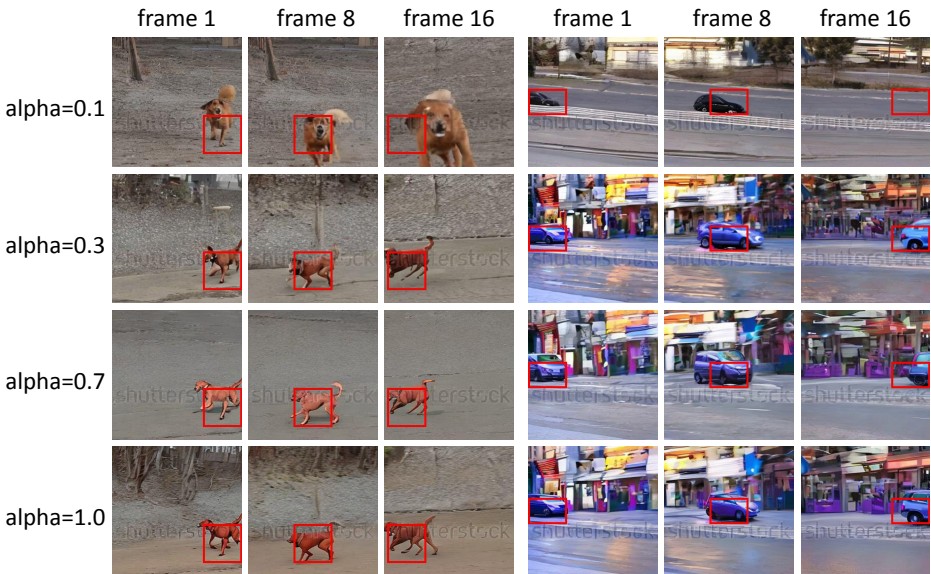

Figure 11: Visualization of objects with different strength of layout control (different $\alpha$ values). The prompts are "A dog moving from right to left" and "A car moving from left to right" respectively. Increasing the $\alpha$ value can make the objects better following bounding box layouts. An $\alpha$ value between 0.1 to 0.3 usually gives videos with best trade-off between visual quality and layout control.

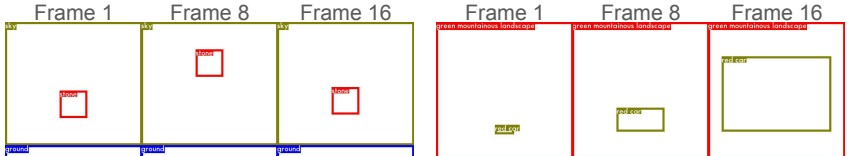

Figure 12: Generated layouts from prompts that require physical understanding: "A stone thrown into the sky" (left) and "A car is approaching from a distance" (right).

et al. (2023). By default, we use $\alpha = 0.1$ and $N = 50$ denoising steps. We also explore using the LLM to determine the $\alpha$ value within the range $[0, 0.3]$ during the *video plan* generation (see main paper Sec. 5.3 for details).

**Visualization and Analysis.** Here, we provide visualizations of objects with prompts that involve explicit layout movements to illustrate the effect of changing the $\alpha$ hyper-parameter. As can be seen in Fig. 11, a small $\alpha$ value (*e.g.*, 0.1) can provide **guidance** for object movement, enabling the Layout2Vid to generate videos that do not strictly adhere to bounding box layouts. This approach fosters better creativity and diversity in the generated videos, and enhances visual quality as well as robustness to LLM-produced layouts. On the other hand, an $\alpha$ value of 0.3 or higher is sufficient to **control** objects to follow bounding box trajectories. In conclusion, for prompts requiring explicit movement control, a larger $\alpha$ value typically results in better performance (as demonstrated by the results from ActionBench-Directions in Table 4 of our main paper). Meanwhile, for prompts that do not require large movements, a smaller $\alpha$ value generally leads to improved visual quality (as indicated by the results from MSR-VTT in Table 4 in the main paper).

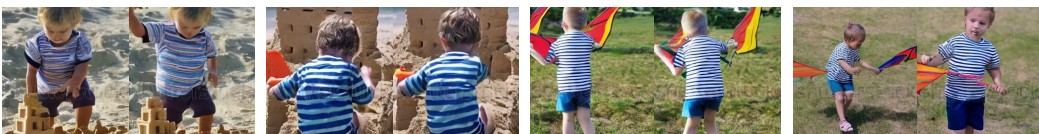

Figure 13: Generated videos from movement prompts with a static object (bottle) and objects that can naturally move (airplan/boat). We use prompts "A {bottle/airplane/boat} moving from left to right."

Figure 14: Multi-scene video generation example from prompt: "A boy playing in sand and flying kites."

## J.2 Additional Qualitative Examples

**Prompts requiring understanding of physical world.** Fig. 12 shows that our GPT-4 based video planner can generate object movements requiring an understanding of the physical world such as gravity and perspective.

**Object movement prompts with different types of objects.** In Fig. 13, we show additional examples for single-scene video generation with prompts involve movements. For static objects (*e.g.*, a bottle), movement is often depicted via the camera. For objects that can naturally move (*e.g.*, an airplane and a boat), the videos show the movements of objects.

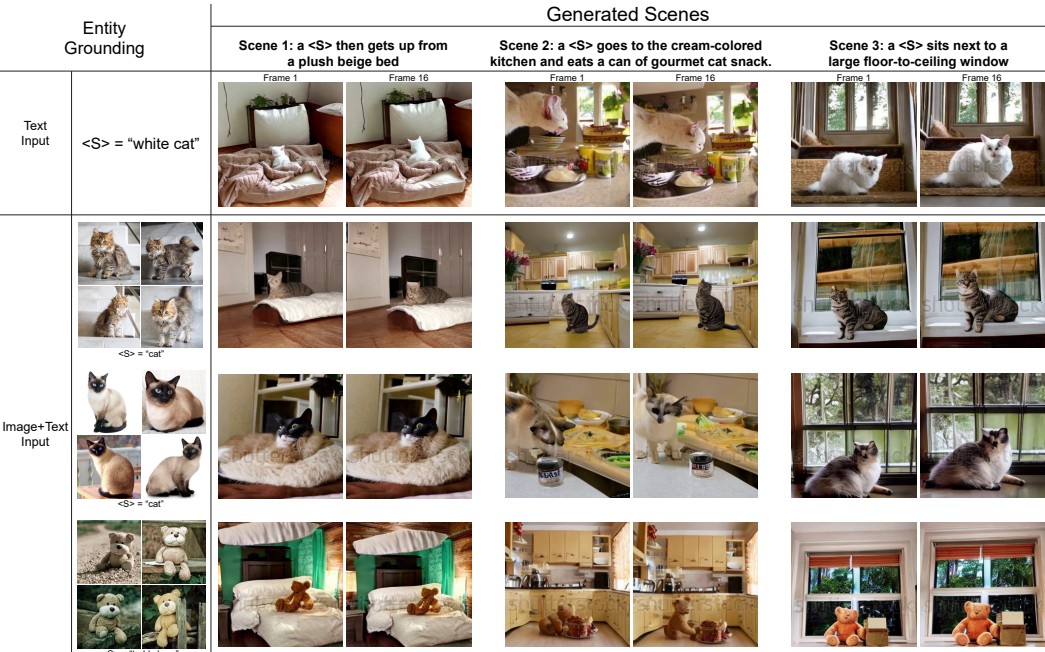

Figure 15: Video generation examples with **text-only and image+text inputs**. Users can flexibly provide either text-only or image+text descriptions to place custom entities when generating videos with VIDEODIRECTORGPT. For both text and image+text based entity grounding examples, the identities of the provided entities are well preserved across multiple scenes.

**Generating videos with custom image exemplars.** In Fig. 15, we demonstrate that our Layout2Vid can flexibly take either text-only or image+text descriptions as input to generate multi-scene videos with good entity consistency.

**Multi-scene video generation with human appearance.** In Fig. 14, we show an additional example of multi-scene video generation with a human appearance. Our VIDEODIREC-TORGPT can well preserve the boy's appearances across scenes.

**VPEval Skill-based.** Fig. 16 shows that LLM-generated video plan successfully guides the Layout2Vid module to accurately place objects in the correct spatial relations and to generate the correct number of objects. In contrast, ModelScopeT2V fails to generate a 'pizza' in the first example and overproduces the number of frisbees in the second example.

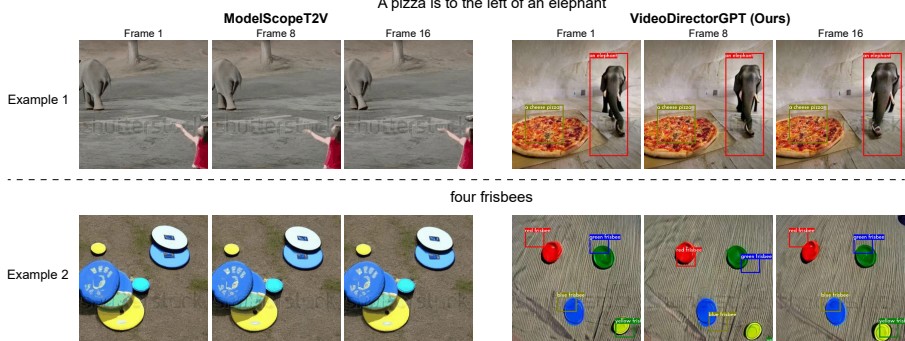

Figure 16: Video generation examples on **VPEval Skill-based prompts** for spatial and count skills. Our *video plan*, with object layouts overlaid, successfully guides the Layout2Vid module to place objects in the correct spatial relations and to depict the correct number of objects, whereas ModelScopeT2V fails to generate a 'pizza' in the first example and overproduces the number of frisbees in the second example.

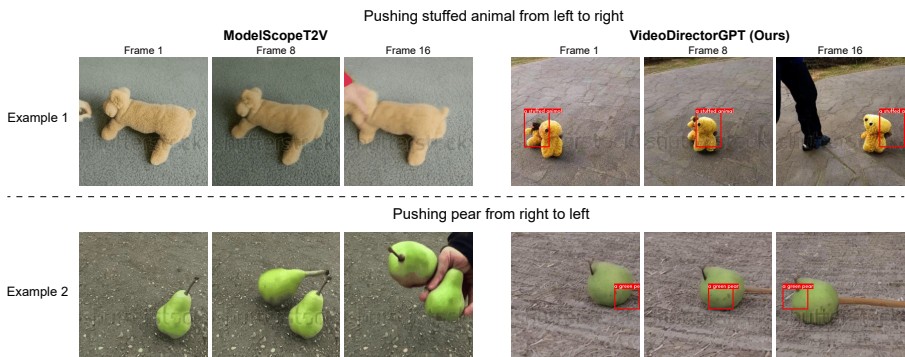

Figure 17: Video generation examples on **ActionBench-Direction prompts**. Our *video plan*'s object layouts (overlaid) can guide the Layout2Vid module to place and move the 'stuffed animal' and 'pear' in their correct respective directions, whereas the objects in the ModelScopeT2V videos stay in the same location or move in random directions.

**ActionBench-direction.** Fig. 17 shows that our LLM-generated video plan can guide the Layout2Vid module to place the 'stuffed animal' and the 'pear' in their correct starting positions and then move them toward the correct end positions, whereas the objects in the ModelScopeT2V videos stay in the same location or move in random directions.

**Coref-SV.** Fig. 18 shows that our *video plan* guide the Layout2Vid module to generate the same dog and maintain snow across scenes consistently, whereas ModelScopeT2V generates different dogs in different scenes and loses the snow after the first scene.

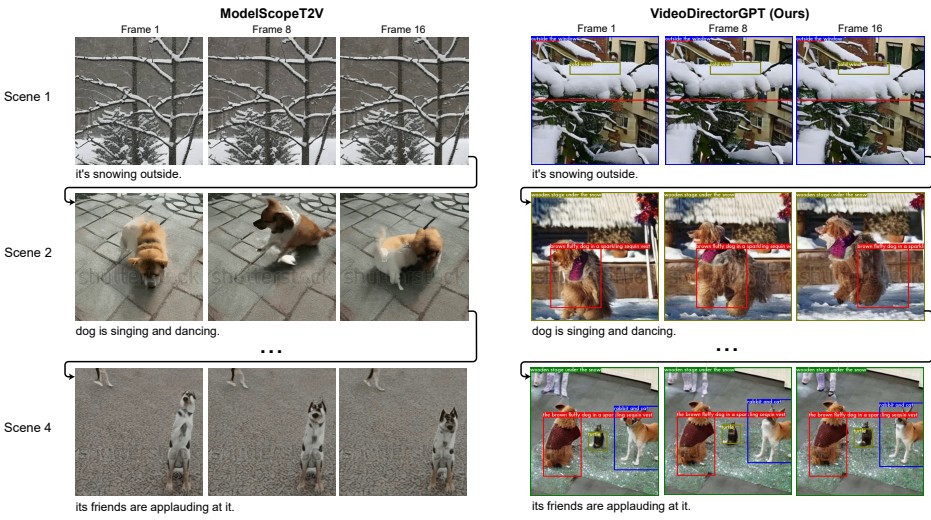

Figure 18: Video generation examples on **Coref-SV prompts**. Our *video plan*'s object layouts (overlaid) can guide the Layout2Vid module to generate the same brown dog and maintain snow across scenes consistently, whereas ModelScopeT2V generates different dogs in different scenes and loses the snow after the first scene.

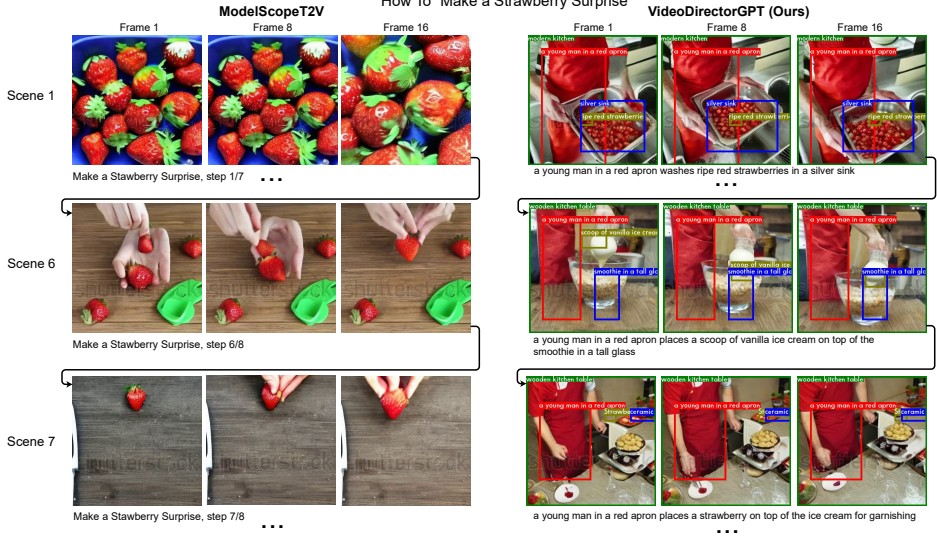

Figure 19: Video generation examples **HiREST prompts**. Our VIDEODIRECTORGPT generates a detailed *video plan* that properly expands the original text prompt, ensures accurate object bounding box locations (overlaid), and maintains consistency of the person across the scenes. ModelScopeT2V only generates strawberries, without strawberry surprise.

**HiREST.** Fig. 19 shows that our LLM can generate step-by-step *video plan* from a single prompt, and our Layout2Vid can generate consistent videos following the plan. Our VIDEODIRECTORGPT breaks down the process and generates a complete video showing how to make a strawberry surprise (a type of dessert consisting of vanilla ice cream and strawberries). ModelScopeT2V repeatedly generates strawberries.

# K   Limitations

While our framework can be beneficial for numerous applications (*e.g.*, user-controlled/human-in-the-loop video generation/manipulation and data augmentation), akin to other video generation frameworks, it can also be utilized for potentially harmful purposes (*e.g.*, creating false information or misleading videos), and thus should be used with caution in real-world applications (with human supervision, *e.g.*, as described in Appendix E — human-in-the-loop video plan editing). Also, generating a *video plan* using the strongest LLM APIs can be costly, similar to other recent LLM-based frameworks. We hope that advances in quantization/distillation and open-source models will continue to lower the inference cost of LLMs. Lastly, our video generation module (Layout2Vid) is based on the pretrained weights of a specific T2V generation backbone. Therefore, we face similar limitations to their model, including deviations related to the distribution of training datasets, imperfect generation quality, and only understanding the English corpus.

