# OpenReview forum: "VideoDirectorGPT: Consistent Multi-Scene Video Generation via LLM-Guided Planning"
_colmweb.org/COLM/2024/Conference — COLM_

### Official Review · Reviewer_Ycer · 2024-05-09

**Rating:** 5
**Confidence:** 3
**Ethics Flag:** 1

**Summary:**

This paper presents VideoDirectorGPT, a multi-scene video generation framework focusing on generation consistency, which leverages the knowledge embedded in LLMs for video content planning and grounded video generation. VideoDirectorGPT consists of two stages. In the first stage, given a single text prompt, GPT-4 is employed as a video planner to craft a video plan, a multi-component video script with multiple scenes to guide the downstream video synthesis process. In the second stage, a novel grounded video generation module named Layout2Vid is proposed to render videos based on the generated video plan. Layout2Vid is built upon the off-the-shelf T2V generation model ModelScopeT2V, thus enabling Layout2Vid to be trained solely on layout-annotated images. Experiments demonstrate that VideoDirectorGPT can generate high-quality videos with good consistency.

**Reasons To Accept:**

The idea of using LLMs as a video plan generator is straightforward and technically sound, and empirically works quite well.

**Reasons To Reject:**

- VideoDirectorGPT is only compared against ModelScopeT2V. The authors should also benchmark other state-of-the-art baseline alternatives such as Phenaki.

- The technical contribution of the paper is limited since the main video generation module, Layout2Vid, is built upon the off-the-shelf T2V generation model ModelScopeT2V.

- Only 5 crowd-annotators from AMT were used, which makes the human evaluation results less reliable.

---

> ### Author Rebuttal · Authors · 2024-05-30
>
> Thank you for your valuable feedback!
>
> **W1: Comparison with more baselines**
>
> As requested, we include **both qualitative and quantitative** comparisons with additional baselines.
>
> Firstly, we compare VideoDirectorGPT quantitatively on layout and object movement control abilities with three more strong baselines in video generation - AnimateDiff, I2VGen-XL, and SVD. The table columns below are same as Table 1 in the main paper. Among the new baselines, I2VGen-XL achieves the highest overall accuracy, while VideoDirectorGPT still outperforms all the baselines including I2VGen-XL by a large margin.
>
> ||||||||
> |-|-|-|-|-|-|-|
> |Method|VPEval Object|VPEval Count|VPEval Spatial|VPEval Scale|VPEval Overall Acc. (%)|ActionBench Movement Direction Acc (%)|
> |ModelScopeT2V|89.8|38.8|18.0|15.8|40.8|30.5|
> |AnimateDiff|96.7|52.7|22.5|15.5|46.8|29.0|
> |I2VGen-XL|96.5|62.0|35.2|23.7|54.3|35.2|
> |SVD|93.1|46.7|29.2|15.0|45.7|30.7|
> |VideoDirectorGPT (ours)|**97.1**|**77.4**|**61.1**|**47.0**|**70.6**|**46.5**|
>
> Secondly, since Phenaki does not released official code, we compare our multi-scene video generation results qualitatively with Phenaki's. The generated gifs from our VideoDirectorGPT are under folder **./Phenaki Prompts** in this link: https://anonymous.4open.science/r/VideoDirectorGPT_COLM-CDC8. Compared with Phenaki, our model is able to generate objects (astronaut, teddy bear) with comparable consistency while equipped with additional layout control.
>
>
> **W2: Technical novelty**
>
> Please see our response to reviewer **ts3J**’s **W2** for more details.
>
>
> **W3: Number of human annotators**
>
> Thanks for this question! We are happy here to further increase the number of human evaluators from 5 to 10. We listed our previous result with 5 human annotators and the new result with 10 (previous 5 + new 5) annotators below.
>
> As we can see, except for the percentage of Ties for Text-Video Alignment, which decreases from 24% to 6%, the relative preference between the two models in the three criteria remains very similar. In both results, our VideoDirectorGPT consistently outperforms ModelScopeT2V in all metrics.
>
>
> **Previous** result with **5** human annotators
> |||||
> |-|-|-|-|
> ||Ours|ModelScopeT2V|Tie|
> |Text-Video Alignment|**54**|22|24|
> |Quality|**52**|30|18|
> |Object Consistency|**62**|28|10|
>
>
> **New** result with **10** human annotators
> |||||
> |-|-|-|-|
> ||Ours|ModelScopeT2V|Tie|
> |Text-Video Alignment|**58**|36|6|
> |Quality|**50**|34|16|
> |Object Consistency|**62**|28|10|

---

> ### Author Response · Authors · 2024-06-04
> **A Gentle Reminder for Response**
>
> We thank the reviewer for their time and effort in reviewing our paper.
>
> We hope that our response has addressed all the questions and hope that the reviewer can consider revising the score based on our response. We are also happy to discuss any additional questions.
>
> With sincere regards,
>
> The authors

---

> ### Comment · Reviewer_Ycer · 2024-06-05
>
> I would like to thank the authors for providing additional qualitative and quantitative results. After reviewing the authors' rebuttal and other reviewers' comments, I am keeping my original score.
>
> I have a similar concern with reviewer ts3J about the novelty and technical contribution of the paper. The proposed model is built on top of the off-the-shelf T2V generation model ModelScopeT2V, and the technical contribution is thus limited. Increasing the number of human annotators from 5 to 10 is useful. However, in my opinion, it is still too small a sample size to be reliable.

---

> > ### Author Response · Authors · 2024-06-05
> > **Response to Reviewer Ycer's Comment**
> >
> > Thanks for your questions. We provide response and clarification as below:
> >
> > >1. The proposed model is built on top of the off-the-shelf T2V generation model ModelScopeT2V, and the technical contribution is thus limited.
> >
> > - We would like to respectfully disagree with the statement that building on top of T2V generation foundation models could lead to the claim that technical contribution is limited. As an analogy, many LLM researchers usually build their methods on top of existing language models (e.g., Llama) instead of training such foundation models from scratch. Similarly, the training cost of T2V generation models like ModelScopeT2V is too high to be affordable for most of the researchers. In addition, our method is model agnostic/easy to adapt to other video generation backbones. Therefore, we kindly request the reviewer to reconsider this statement.
> >
> >
> > >2. Increasing the number of human evaluators from 5 to 10 is useful. However, in my opinion, it is still too small a sample size to be reliable.
> >
> > - We would like to bring to the reviewer’s attention that using less than 10 or around 10 human evaluators is very common in related works in NLP [1], multi-modal AI [2], diffusion models [3], and LLM+diffusion works [4, 5], and these papers are well received at top conferences and journals such as EMNLP, ICLR, ICCV, and TMLR. These works include ControlNet [3], which won the best paper award in ICCV 2023, and MaRVL [2], the best paper in EMNLP 2021.
> >
> >
> > We hope our clarification is helpful. Please let us know if you have any other questions. Thanks!
> >
> >
> > [1] Cho, Sangwoo, et al. "Toward Unifying Text Segmentation and Long Document Summarization." Proceedings of the 2022 Conference on Empirical Methods in Natural Language Processing. 2022.
> >
> > [2] Liu, Fangyu, et al. "Visually Grounded Reasoning across Languages and Cultures." Proceedings of the 2021 Conference on Empirical Methods in Natural Language Processing. 2021.
> >
> > [3] Zhang, Lvmin, Anyi Rao, and Maneesh Agrawala. "Adding conditional control to text-to-image diffusion models." Proceedings of the IEEE/CVF International Conference on Computer Vision. 2023.
> >
> > [4] Lian, Long, et al. "LLM-grounded Video Diffusion Models." The Twelfth International Conference on Learning Representations. 2023.
> >
> > [5] Lian, Long, et al. “LLM-grounded Diffusion: Enhancing Prompt Understanding of Text-to-Image Diffusion Models with Large Language Models.” Transactions on Machine Learning Research (TMLR)

---

### Official Review · Reviewer_tgxZ · 2024-05-11

**Rating:** 7
**Confidence:** 2
**Ethics Flag:** 1

**Summary:**

The authors proposed a two-stage text-to-video pipeline, VideoDirectorGPT, with two major components: (1) a video planner and (2) a video generator. The video planner takes a video text prompt to generate video plans using GPT4, which also groups the same entities across scenes and generates layout (bounding box) of entities in addition to background and scene description. THe video planner is finetuned from ModelScopeT2V for training temporally consistent entity grounding component. Note that only 13% of total model weights are updated for this component.

Experiment results show that the proposed VideoDirectorGPT outperforms ModelScopeT2V in single-scene and multi-scene settings on both automatical and human evaluation. Overall, the authors did a great job evaluating the video generator component.

**Questions To Authors:**

Comments/Suggestions:
It would be helpful to have an example of full video planner output in the appendix.

**Reasons To Accept:**

1. Comprehensive evaluation of generated video generator, including automatic evaluation and human evaluation with ablation studies.
2. Proposed method can take not only textual input but also image + text input.
3. The two-stage pipeline allows users to modify each of the components, such as background or bounding boxes.

**Reasons To Reject:**

Not that I can think of. Basically, all of my questions are answered in the appendix.

---

> ### Author Rebuttal · Authors · 2024-05-30
>
> Thank you for valuing our comprehensive evaluation, the technical novelty of our proposed method, and our user-friendly two-stage framework design!
>
> As suggested, we will include examples of full video planner output in the appendix in our camera-ready version. Please let us know if you have any other questions or concerns!

---

### Official Review · Reviewer_ts3J · 2024-05-17

**Rating:** 4
**Confidence:** 5
**Ethics Flag:** 1

**Summary:**

This paper proposes VIDEODIRECTORGPT for video generation. It comprises two stages: the first stage is an LLM planner that generates the scene descriptions, the entities with their respective layouts, the background for each scene, and consistency groupings of the entities with the given prompt; the second stage is a video generator mainly based on the weights of ModelScopeT2V. The authors claim their experiments demonstrate that their proposed framework improves layout and movement control in both single- and multi-scene video generation and can generate multi-scene videos with consistency, while achieving competitive performance with SOTAs in open-domain single-scene T2V generation. Ablation studies confirm the effectiveness of each component of the proposed framework.

**Questions To Authors:**

Refer to weaknesses.

The authors' response addresses my first reject reasion, which is appreciated. However, after carefully checking the code, I maintain my reservation regarding the heavy reliance on GLIGEN and ModelScopeT2V, which dominates the described method. Moreover, the emphasis on video rather than LM suggests that this work falls short of the expectations set by the CoLM standards.

**Reasons To Accept:**

1. The paper is well-written and easy to understand.

2. The experiments are sufficient in many aspects, including single-scene and multi-scene video generation, with metrics like FID, FVD, IS, and human evaluations.

**Reasons To Reject:**

1. The authors mentioned that "We provide code and generated video examples in the supplementary material," but I did not find where the supplementary material is. Thus, I am concerned about the completeness of this work, as well as the reproducibility and impact of the proposed method due to the lack of related code and models.

2. The application is not novel at all. It mainly relies on GLIGEN and ModelScopeT2V. One image version work is [1]. Thus, this work is a little bit below the bar of CoLM.

3. For a video generation paper, related videos should be appended to demonstrate the real performance. As we all know, some quantitative evaluation metrics are inconsistent with real performance, as many studies have proved.

4. There is a lack of related ablation experiments on different language models and whether to fine-tune them. At an LLM conference, the authors should focus more on the LM itself instead of video generation.


[1] LLM-grounded Diffusion: Enhancing Prompt Understanding of Text-to-Image Diffusion Models with Large Language Models, https://llm-grounded-diffusion.github.io/

---

> ### Author Rebuttal · Authors · 2024-05-30
>
> Thank you for your valuable feedback!
>
> **W1: Supplementary materials**
>
> We would like to bring your attention to our “supplementary materials” below the abstract. Please feel free to let us and ACs know if you didn't find the supplementary materials in your openreview system.
>
> **W2: Novelty/Contributions**
>
> We would like to clarify that LLM-grounded Diffusion [1] and our paper handle different tasks with different methods. Firstly, [1] only covers T2I generation, while our work proposes a new framework for multi-scene T2V generation. Secondly, [1] is about inference-time optimization, while our method focuses on model training without inference-time optimization.
>
> **In addition, we would like to elaborate our novelty and contributions:**
>
> - **1.** We introduce **the first multi-scene video generation framework by leveraging the LLM knowledge for planning and ensuring consistency across scenes.**
>
> - **2(a) Video Planner:** Our video planner introduces a complex multi-component video plan (scene descriptions, entity descriptions and layouts, and consistency groupings) for multi-scene video generation.
>
> - **2(b) Layout2Vid:** We propose to insert layout control layers (similar to GLIGEN but with enhanced cross-scene text/image entity representation) within frozen T2V backbone, which supports multi-scene consistency control and image-only training, while preserving the knowledge of the T2V model.
>
> - **3. We provide comprehensive qualitative/quantitative experiments** including novel evaluation metrics (movement direction accuracy / entity consistency) / new datasets (ActionBench-Direction/Coref-SV) for video generation, as recognized by all other reviewers.
>
> **W3: Video examples**
>
> We would like to bring your attention to the html file in our supplementary materials, which includes many visualization examples in **.gif** format. Besides, we provide human evaluation in Section 5.4. We also further increased the number of human evaluators from 5 to 10, as suggested by reviewer Ycer (see the response to reviewer **Ycer W3** for details).
>
> **W4: LLM ablation**
>
> We would like to bring your attention to Appendix section H.1, where we have provided detailed ablation experiments on different language models: GPT-4, GPT-3.5-turbo, and LLaMA2.
>
> **Considering that 3 out of the 4 reasons to reject were already addressed in our appendix/supplementary materials, and we have elaborated our novelty/contributions, we kindly request to raise the score accordingly. Thanks!**

---

> > ### Author Response · Authors · 2024-06-04
> > **Response to updated reviewer comment**
> >
> > **Part1 response to updated reviewer comment:**
> > >Reviewer ts3J: "The authors' response addresses my first reject reason, which is appreciated. However, after carefully checking the code, I maintain my reservation regarding the heavy reliance on GLIGEN and ModelScopeT2V, which dominates the described method.”
> >
> >
> > To begin with, we would like to re-state that in addition to the first rejection reason, the third and the fourth rejection reason were already covered in our first rebuttal response.
> > We respectfully disagree with the assertion that our primary contribution relies heavily on GLIGEN and ModelScopeT2V. As we mentioned in the rebuttal, our paper contribution consists of three parts: (1) LLM for video plan generation, (2) Layout-guided video generation with consistency control, as well as (3) Novel evaluation metrics and datasets for measuring layout and consistency accuracy for (multi-scene) video generation, as detailed below:
> >
> > (1) Firstly, we would like to emphasize that a substantial part of our work focuses on introducing the use of LLMs to generate complex, useful plans for multi-scene video generation. We start from the motivation/observation that LLMs (GPT-4) can generate bounding box movements with physical world understanding (Fig. 12), then design a multi-component video plan (Sec. 3.1) via in-context learning (Fig. 7 and Fig. 8). Finally, we conducted thorough analysis of the effect of different LLMs (Table 9), different layout control methods (Table 11), as well as error analysis (Table 8). We believe our paper has covered the analysis of LLMs in a relatively complete way. If you have any other questions or concerns, we are more than willing for open discussion.
> >
> > (2) Secondly, we would like to highlight that our method is agnostic to the backbone model. To make our pipeline work, we need to build on one specific video generation model (e.g., ModelScopeT2V, the best open-source T2V model at the time). However, it's easy to adapt our framework to future video generation backbones. In addition, adopting a gated self-attention module from GLIGEN is only one part of our architecture design. We also made a non-trivial effort to enhance it with text+image embedding conditioning, proposed sharable embedding for multi-scene consistency control, and creatively trained the video generation module in a parameter-efficient way with only images (Sec. 3.2). We also did extensive ablation studies of our model architecture design (Table 4, 7, 10, 12). Therefore, stating that our method "relies heavily on GLIGEN and ModelScopeT2V'' is not well-supported.
> >
> > (3) Thirdly, we provide comprehensive qualitative + quantitative experiments including novel evaluation metrics (movement direction accuracy / entity consistency) / new datasets (ActionBench-Direction/Coref-SV) for video generation, which could be beneficial for future works on layout-guided and multi-scene video generation.
> >
> > ----------------------------
> >
> > **Part2 response to updated reviewer comment:**
> >
> > >Reviewer ts3J: “Moreover, the emphasis on video rather than LM suggests that this work falls short of the expectations set by the CoLM standards."
> >
> > **We strongly believe that our novel framework leveraging LMs for video generation aligns well with CoLM’s interests.** Specifically, our work addresses the topic of “17. LMs on diverse modalities and novel applications,” as outlined in the CoLM call for papers (https://colmweb.org/cfp.html).
> >
> > ```17. LMs on diverse modalities and novel applications: visual LMs, code LMs, math LMs, and so forth, with extra encouragement for less studied modalities or applications such as chemistry, medicine, education, database, and beyond.```
> >
> > Moreover, the CoLM homepage (https://colmweb.org/index.html) mentions:
> >
> > ```CoLM is an academic venue focused on the study of language modeling, broadly defined, with the goal of creating a community of researchers with expertise in different disciplines, focused on understanding, improving, and critiquing the development of LM technology.```
> >
> > **These underline the relevance of our contribution to the CoLM community.**
> > **We kindly ask for a reconsideration of the review and rating in light of these points.**

---

> ### Comment · Reviewer_ts3J · 2024-06-05
>
> After carefully reviewing the authors' response and considering other reviewers' comments, I have decided to maintain my rating at [4. rejection] for the following reasons:
>
> 1. **Heavy reliance on GLIGEN and ModelScopeT2V:** Despite the authors' claims of significant contributions, the code indicates a primary dependence on these existing models or methods.
> 2. **Poor organization:** Many important evaluation and ablation studies are relegated to the supplementary materials, which undermines the persuasiveness of the main sections. Given that supplementary materials will not be published, this is a significant issue.
> 3. **Misalignment with conference scope:** As a work focused on video generation, it is not well-suited for a language modeling (LM) conference. Although the authors reference many CoLM criteria, visual LMs are different from diffusion model technologies. This paper primarily centers on video generation with diffusion models rather than LMs. It may be more appropriate for the authors to reorganize their work and submit it to a vision-focused conference.

---

> > ### Author Response · Authors · 2024-06-05
> > **Response to Reviewer ts3J's Comment**
> >
> > >1. Heavy reliance on GLIGEN and ModelScopeT2V: Despite the authors' claims of significant contributions, the code indicates a primary dependence on these existing models or methods.
> >
> > - As we already clearly elaborated in our previous responses, our method is agnostic to the backbone model. To make our pipeline work for the purpose of experimentation, we need to choose one specific video generation model to build upon (e.g., ModelScopeT2V, the best open-source T2V model at the time), and it's easy to adapt our framework to future video generation backbones.
> > As an analogy, LLM researchers usually build their methods on top of existing language models (e.g., Llama), which also indicates a primary dependence on these models. Thus, we believe it’s unfair to criticize a paper due to the foundation model it uses.
> >
> >
> >
> > >2. Poor organization: Many important evaluation and ablation studies are relegated to the supplementary materials, which undermines the persuasiveness of the main sections. Given that supplementary materials will not be published, this is a significant issue.
> >
> > - Firstly, our paper contains thorough analysis of how to use LLMs to help video generation, so we put the most important results in the main paper, and relatively less important ablation studies in the appendix (similar to all other papers submitted to COLM and all other conferences with 8-10 page limits). If the reviewer believes that it’s worthwhile to bring some ablation studies from the appendix back to the main paper, we are very willing to use the extra page given in the camera-ready version for this purpose.
> >
> > - Secondly, **we have emailed COLM PCs last night, and they told us to share the clear confirmation that the supplementary material will be made public along with the paper.**
> >
> > Considering all these aspects, we strongly believe that this should not be a “significant issue”.
> >
> >
> > >3. Misalignment with conference scope: As a work focused on video generation, it is not well-suited for a language modeling (LM) conference. Although the authors reference many CoLM criteria, visual LMs are different from diffusion model technologies. This paper primarily centers on video generation with diffusion models rather than LMs. It may be more appropriate for the authors to reorganize their work and submit it to a vision-focused conference.
> >
> >
> > - As we already mentioned in our previous responses, our focus is on how to use LLMs to help the video generation task. Using visual LMs could be one technical design choice, and using LLMs+diffusion could be another design choice. We chose our approach because diffusion models are the most popular framework for modern image/video generation currently. Given that both methods are under the topic of using LLMs to help video generation, we strongly believe that the specific technical design choices to tackle this problem should not be the key point to distinguish if a paper should be incorporated into COLM.

---

### Official Review · Reviewer_HYoe · 2024-05-20

**Rating:** 6
**Confidence:** 3
**Ethics Flag:** 1

**Summary:**

This papar propose video-director-GPT, a framework to generate consistent multi-scene videos.
The method includes a GPT-4 planning process and a layout2vid module to generate video based on the plans.
The experiments show that the quality is improved.

**Questions To Authors:**

Please see the comments above.

**Reasons To Accept:**

1. the idea and the method is clear and well-motivated
2. the evalution is strong, including automatic and annoatation evaluations.

**Reasons To Reject:**

1. the experiments are somehow limited, I understand the fact that running LLM servers might be difficult, still, only using GPT-4 as the planner without any other alternative LLMs might weaken the results.
2. If I'm not mistaking any details, the GPT-4 generated plans are later grouped into a shared representation based on exact match. Is this a method that weakens the power of LLMs? The reasoning, continuous semantics are all discarded then?

---

> ### Author Rebuttal · Authors · 2024-05-30
>
> Dear reviewer HYoe, thank you for your valuable feedback! We address your concerns with clarifications as follows.
>
>
> **W1: experiments with other LLMs**:
> We would like to remind you that we have provided detailed ablation experiments on different language models and whether to fine-tune them in the section H.1 Video Planner LLMs: GPT-4, GPT-3.5-turbo, and LLaMA2.
> Please let us know if you have any further questions about ablations of different LLMs, we are very glad to address.
>
>
> **W2: is exact match weakening entity semantics?**
>
> That is a great question! We would like to point out that when we ask GPT-4 to generate entity names, the rich semantics (e.g., object attributes) can be well maintained. For example, as shown in Fig 2 and Fig 5, the entities generated by GPT-4 contain “a fluffy Siamese cat”, “a plush beige bed”, “woman in a green apron”, “modern kitchen counter”, and so on. These texts, instead of just “cat”, “bed”, “woman”, and “kitchen”, are used to construct object embeddings that are shared across all scenes.
> In addition, if the user wants more fine-grained control over object appearance, they can provide a reference image in addition to text descriptions (as shown in Fig. 6). For example, if the same woman changes from a "woman in a green apron" in scene 1 to a "woman in a red T-shirt" in scene 2, we can make such changes while maintaining the appearance of the woman by editing the reference image from a "green apron" to a "red T-shirt" using off-the-shelf image-editing tools (e.g., Instructpix2pix [1]).
>
>
> [1] Brooks, Tim, Aleksander Holynski, and Alexei A. Efros. "Instructpix2pix: Learning to follow image editing instructions." Proceedings of the IEEE/CVF Conference on Computer Vision and Pattern Recognition. 2023.

---

> ### Author Response · Authors · 2024-06-04
> **A Gentle Reminder for Response**
>
> We thank the reviewer for their time and effort in reviewing our paper.
>
> We hope that our response has addressed all the questions and hope that the reviewer can consider revising the score based on our response. We are also happy to discuss any additional questions.
>
> With sincere regards,
>
> The authors

---

> > ### Comment · Reviewer_HYoe · 2024-06-05
> >
> > I'm curious what is the random accuracy in ActionBench-directions? I'm not sure if llama models can handle such task? I believe I've asked this question above

---

> > > ### Author Response · Authors · 2024-06-05
> > > **Response to Reviewer HYoe's Comment**
> > >
> > > Dear reviewer HYoe, thanks for your questions. We provide response and clarification as below:
> > >
> > > >I'm not sure if llama models can handle such task?
> > >
> > > - Thanks, we would like to bring your attention to our previous response to W1 (https://openreview.net/forum?id=sKNIjS2brr&noteId=UaqWRftSIi), we have shown the comparison between LLaMA2, GPT-3.5-turbo, and GPT-4 in Table 9 of Section H.1.
> > >
> > > - From the result in Table 9, we can see that fine-tuned LLaMA2-13B performs in-between GPT-3.5-turbo and GPT-4 (GPT-3.5-turbo 49.0 v.s. LLaMA2-13B 53.3 v.s. GPT-4 59.8).
> > >
> > >
> > > >I'm curious what is the random accuracy in ActionBench-directions?
> > >
> > > - As described in the paragraph “ActionBench-Direction prompts” on page 20 of our paper, for each object, we give prompts that include four movement directions: ‘left to right’, ‘right to left’, ‘top to bottom’, and ‘bottom to top’. Therefore, for a random guess, the movement accuracy on the ActionBench-directions dataset should be around 25%.
> > >
> > >
> > > We hope our clarification is helpful. Please let us know if any part of our response is still unclear to you. Thanks!

---

> > > > ### Comment · Reviewer_HYoe · 2024-06-07
> > > >
> > > > Thanks for the answer.

---

### Official Review · Reviewer_TCNG · 2024-05-23

**Rating:** 6
**Confidence:** 5
**Ethics Flag:** 1

**Summary:**

This paper introduces VideoDirectorGPT, which leverages large language models (LLMs) to derive object/layout plans for multi-scene long video generation. Specifically, the video planner expands the input caption into scene descriptions, including entities with bounding boxes and the overall background. Then, the proposed Layout2Vid can control explicit spatial layout and temporal coherence during video generation.

**Questions To Authors:**

Please see Reasons To Reject

**Reasons To Accept:**

- This paper is well-written and easy to follow.
- The idea of adopting LLMs to derive scene description/layout is well-motivated, which has the potential to benefit the planning of long video generation.
- They provide detailed ablation studies and comprehensive visualized examples in the Appendix.

**Reasons To Reject:**

- Since there are several previous works [1,2] adopting LLMs to improve the layout planning of T2I, the novelty of this project can be the issue. Furthermore, the proposed Layout2Vid is an extension of the existing GLIGEN [3]. I am afraid that this project can not achieve the bar of a top-tier conference.
- Though the proposed video planner can craft the plan, Layout2Vid only generates 16 frames (2-4 seconds) as the output video. Can this short clip actually contain multiple scenes (e.g., different entities and backgrounds)? I am worried about the practicality of the proposed framework.
- Apart from those successful cases, failure examples are also crucial to discuss, which can guide future research on this topic.

**Reference**
- [1] LayoutGPT: Compositional Visual Planning and Generation with Large Language Models
- [2] LLM-grounded Diffusion: Enhancing Prompt Understanding of Text-to-Image Diffusion Models with Large Language Models
- [3] GLIGEN: Open-Set Grounded Text-to-Image Generation

---

> ### Author Rebuttal · Authors · 2024-05-30
>
> Thanks for your valuable feedback!
>
> **W1: Our novelty, and comparison with GLIGEN**
>
> Please see our response to reviewer **ts3J**’s **W2** for more details.
>
> **W2: Is the multi-scene video only constrained to 16 frames?**
>
> We would like to clarify that the generated multi-scene video consists of multiple single-scene videos, while each single-scene video contains 16 frames. The video plan can contain as many scenes as is needed for the video.
>
> In addition, the number of frames in each single scene (16 here) are dependent on the video generation model (ie, ModelScopeT2V), which can be easily extended with backbones that can generate more frames (25 frames in SVD, and even 221 frames in open-sora-plan). Some training-free frame-extension methods (e.g., FreeNoise [1]) can be seamlessly integrated into our framework.
>
> **W3: Failure cases**
>
> Our method inherits two shortcomings of bounding-box-based control.
>
> Firstly, some objects might not follow the bounding box control well when there are too many overlapping bounding boxes. When multiple objects are in the same overlapping bounding box area, the CLIP image embeddings from the cropped object regions share pixels, making it difficult to disentangle their identities. As we can see from the gifs under the folder **./Failure Cases** in this link: https://anonymous.4open.science/r/VideoDirectorGPT_COLM-CDC8. The “fresh lemon” and “sliced onion”, which have relatively small bounding box sizes, are not following the box control well. Adopting layer-wise object generation (e.g., LayerDiffuse [2]) could be a potential solution to this problem.
>
> Secondly, when a bounding box does not move across frames (e.g., box of background), it is hard to make sure the background is kept static. As shown in Fig. 13, it’s hard for us to generate a moving bottle with a static background, and a moving boat with a completely static background. The static/dynamic of the background are usually determined by the prior knowledge of the video generation model. Adopting methods that can explicitly handle camera movement (e.g., CameraCtrl [3]) could be a potential solution.
>
> [1] Qiu, et al. "Freenoise: Tuning-free longer video diffusion via noise rescheduling." arXiv preprint arXiv:2310.15169 (2023)
>
> [2] Zhang, et al. "Transparent Image Layer Diffusion using Latent Transparency." arXiv preprint arXiv:2402.17113 (2024)
>
> [3] He, et al. "Cameractrl: Enabling camera control for text-to-video generation." arXiv preprint arXiv:2404.02101 (2024)

---

> > ### Comment · Reviewer_TCNG · 2024-06-05
> >
> > This rebuttal mostly addresses my concerns. I keep my score as 6.

---

> ### Author Response · Authors · 2024-06-04
> **A Gentle Reminder for Response**
>
> We thank the reviewer for their time and effort in reviewing our paper.
>
> We hope that our response has addressed all the questions and hope that the reviewer can consider revising the score based on our response. We are also happy to discuss any additional questions.
>
> With sincere regards,
>
> The authors

---

### Decision · Program_Chairs · 2024-07-10

**Decision:**

Accept

**Comment:**

The paper proposes an approach for planning video scenes using LLMs. Reviewers had several concerns regarding novelty and evaluation. The authors provided results for more baselines and evaluations. Overall I agree there are potentially interesting ideas here, and I propose that the authors address the comments raised.